# MINI-BATCH CORESETS FOR MEMORY-EFFICIENT LANGUAGE MODEL TRAINING ON DATA MIXTURES

**Dang Nguyen**    **Wenhan Yang**    **Rathul Anand**    **Yu Yang**    **Baharan Mirzasoleiman**

{dangnth, hangeryang18, rathul, yuyang, baharan}@cs.ucla.edu
Computer Science Department, UCLA

## ABSTRACT

Training with larger mini-batches improves the convergence rate and can yield superior performance. However, training with large mini-batches becomes prohibitive for Large Language Models (LLMs), due to the large GPU memory requirement. To address this problem, an effective approach is finding small mini-batch coresets that closely match the gradient of larger mini-batches. However, this approach becomes infeasible and ineffective for LLMs, due to the highly imbalanced mixture of sources in language data, use of the Adam optimizer, and the very large gradient dimensionality of LLMs. In this work, we address the above challenges by proposing *Coresets for Training LLMs* (CoLM). First, we show that mini-batch coresets found by gradient matching do not contain representative examples of the small sources w.h.p., and thus including all examples of the small sources in the mini-batch coresets is crucial for optimal performance. Second, we normalize the gradients by their historical exponential to find mini-batch coresets for training with Adam. Finally, we leverage zeroth-order methods to find smooth gradient of the last $V$-projection matrix and sparsify it to keep the dimensions with the largest normalized gradient magnitude. We apply CoLM to fine-tuning Phi-2, Phi-3, Zephyr, and Llama-3 models with LoRA on MathInstruct and SuperGLUE benchmark. Remarkably, CoLM reduces the memory requirement of fine-tuning by 2x and even outperforms training with 4x larger mini-batches. Moreover, CoLM seamlessly integrates with existing memory-efficient training methods like LoRA, further reducing the memory requirements of training LLMs.

## 1 INTRODUCTION

Large Language Models (LLMs) have achieved remarkable success in a variety of tasks, ranging from machine translation to conversational AI. However, pretraining and fine-tuning LLMs with billions of parameters requires a large amount of compute and GPU memory, not only to store the parameters but also to compute gradients and optimizer states (e.g., momentum and historical gradients in Adam). For example, full finetuning a relatively small LLM, such as Phi-2 with 2.7B parameters, using a batch size of 128 requires at least 44 GB of GPU memory. The large memory requirement makes it prohibitive to train such models with larger batch sizes, which effectively improves the convergence and can improve performance. This raises a key question: *can we train LLMs with smaller mini-batches and get the benefits of training with larger batch sizes?*

To address this problem, many memory-efficient techniques have been recently proposed, mainly to enable efficient fine-tuning of pretrained language models. At a high level, such methods aim to find a smaller set of parameters (Adelman et al., 2021), or find low-rank (Hu et al., 2021; Zhao et al., 2024b) or quantized (Dettmers et al., 2022) weights or optimizer states to train the model in a memory-efficient manner. There have also been efforts to adapt gradient-free optimization for training LLMs (Malladi et al., 2023). Yet, most memory-efficient techniques cannot achieve a comparable performance to training the full model parameters, or considerably increase the training time.

In this work, we address the above problem from the *data* perspective. Specifically, we target finding smaller mini-batches of examples that simulate or outperform training with larger mini-batches. If this can be done, it directly improves the convergence rate of training or fine-tuning with mini-batch stochastic gradient methods, and can yield superior performance. To achieve this, an effective

approach is to find smaller mini-batches that closely capture the gradient of large random batches (Yang et al., 2023). Smaller mini-batches (coresets) selected by this approach are medoids (centroids) of the data in gradient space, weighted by their corresponding cluster size (Mirzasoleiman et al., 2020). Despite its promise on classification tasks, this approach is infeasible and ineffective for pre-training or fine-tuning LLMs, as we discuss below.

Firstly, language data is often a mixture of highly imbalanced sources (e.g. categories or types of instructions). In this case, we show that smaller mini-batch coresets do not contain representative examples from the small sources, and obtain poor performance. Secondly, Adam is the standard optimizer for training LLMs, and small mini-batches that capture the vanilla gradient of larger batches are not optimal for training with Adam. Finally, the very large dimensionality of the LLM gradients makes pairwise distances vacuous and the medoids cannot be found accurately. These challenges make finding mini-batch coresets for training LLMs inherently much more challenging to address.

In this work, we propose *Coresets for Training LLMs* (CoLM) by making the following contributions:

- First, we show that w.h.p. mini-batch coresets only contain medoids of the big sources with a large-enough number of samples. Thus, examples selected via gradient matching from small sources are not representative and do not benefit learning other examples in their sources. Therefore, it is crucial to include *all* examples of the small sources in the mini-batch coresets. Besides, to enhance learning small sources, we weight all examples in the mini-batch coresets uniformly.

- Next, to find mini-batch coresets for training with Adam, we normalize gradients by their historical exponential average, where the historical terms are only calculated for examples in the big sources. We find medoids of the big sources based on their normalized gradients.

- Finally, to enable finding medoids in the very high-dimensional gradient space, we use zeroth-order methods to find smooth gradient of the last $V$-projection matrix in a memory-efficient way, and sparsify it via a source-wise mask, which keeps dimensions with the largest normalized gradient magnitude. We use $\ell_1$ distance between sparse gradients to find medoids of big sources.

- We evaluate CoLM on the challenging task of mathematical problem-solving, by fine-tuning Phi-2, Phi-3 (Li et al., 2023b), Zephyr (Tunstall et al., 2023), and Llama-3 models (Dubey et al., 2024) using LoRA on the MathInstruct dataset (Yue et al., 2023) containing 14 highly imbalanced sources. Additionally, we apply CoLM to datasets in SuperGLUE benchmark (Wang et al., 2019), where we find sources by clustering the model's hidden states during the training. Remarkably, CoLM reduces the memory requirement of fine-tuning by 2x and even outperforms training with 4x larger random mini-batches. Compare to mini-batch of the same size, CoLM outperforms by up to 7.1% and 20% on several in- and out-domain tasks.

Notably, our approach can be easily stacked with LoRA, and other memory-efficient training methods to further reduce the memory requirements of training LLMs, as we confirm in our experiments.

## 2 RELATED WORK

**Memory-efficient training of LLMs.** To address the large memory requirements of training LLMs, several methods have been recently proposed. LoRA (Hu et al., 2021) freezes the pre-trained model weights and trains two low-rank adaptor weight matrices to adapt the weights of each layer. However, LoRA suffers from a performance drop compared to training with full-rank matrices. To improve upon this, several variations of LoRA (Liu et al., 2024; Renduchintala et al., 2023; Xia et al., 2024b) have been proposed. Besides, GaLore (Zhao et al., 2024b) proposed to reduce the memory cost of optimizer states by calculating the gradients and projecting them into a low-rank space. However, the above approaches also lead to increased computational costs.

Another line of methods approximate backpropagation by sparsifying gradients (Frantar and Alistarh, 2023), subsampling the computational graph (Adelman et al., 2021), gradient check-pointing (Chen et al., 2016), and quantization of weights and optimizer states (Dettmers et al., 2022). However, these approaches can incur large approximation errors and cause performance drops. Zeroth-order gradient approximation has also been used for memory-efficient training (Malladi et al., 2023). However, this approach cannot reach a comparable performance to normal training.

Our method can be easily stacked with existing memory-efficient methods to improve convergence and further reduce memory requirements.

**Data selection for training LLMs.** Data selection for training LLMs has garnered significant attention due to its potential to enhance model performance while reducing computational costs. For pre-training, examples with middle perplexity rankings are shown beneficial (Marion et al., 2023). Clustering based on embeddings of a pretrained model and sampling from the clusters to drop redundancies has been also investigated (Tirumala et al., 2024).

For fine-tuning, training on manually crafted high-quality instruction/response pairs has shown highly effective (Zhou et al., 2023a). Building on this observation, data selection using LLMs such as ChatGPT or training on textbooks is proposed (Eldan and Li, 2023; Li et al., 2023c; Chen et al., 2024; Li et al., 2023a), and metrics such as diversity (Bukharin and Zhao, 2023; Du et al., 2023), difficulty (Bhatt et al., 2024; Marion et al., 2023; Zhou et al., 2023b), and completion length (Zhao et al., 2024a) are shown relevant. Given a high-quality validation set, using influence functions to select the most beneficial subsets of fine-tuning data has been also explored (Xia et al., 2024a). However, a high-quality validation set is not always available (e.g. for MathInstruct). Existing methods select data in a one-shot manner before fine-tuning, and either require access to another open LLM or a large preprocessing time to fine-tune the original or a proxy LLM on the target data.

We study data selection from a different perspective, i.e. by selecting small mini-batches that match the performance of training with larger mini-batches. As baselines, we adapt several one-shot methods based on loss (Jiang et al., 2019), gradient norm (Katharopoulos and Fleuret, 2018), middle perplexity (Marion et al., 2023), completion length (Zhao et al., 2024a), confidence, and hidden-state centrality (Bhatt et al., 2024) to iteratively select small mini-batches from larger random batches.

## 3 PRELIMINARY: MATCHING GRADIENT OF LARGE BATCHES

Consider training a machine learning model on a dataset indexed by $V$, by minimizing the loss function $\mathcal{L}(\boldsymbol{\theta}) = \mathbb{E}_{i \in V}[\mathcal{L}_i(\boldsymbol{\theta})]$. Mini-batch SGD with learning rate $\eta$ iteratively updates the model parameters as $\boldsymbol{\theta}_{t+1} = \boldsymbol{\theta}_t - \eta \, \mathbf{g}_{\mathcal{M}_t,t}$, where $\mathbf{g}_{\mathcal{M}_t,t} = \mathbb{E}_{i \in \mathcal{M}_t}[\mathbf{g}_{i,t}]$ is the gradient of a mini-batch $\mathcal{M}_t$ of random examples, and $\mathbf{g}_{i,t} = \nabla \mathcal{L}_i(\boldsymbol{\theta}_t)$. As long as the mini-batch size $b = |\mathcal{M}_t|$ is not too large, the convergence rate of mini-batch SGD directly scales with a factor of $1/b$. Formally, for a non-convex $L$-gradient Lipschitz loss, mini-batch SGD with a small enough $\eta$ will visit an $\epsilon$-stationary point w.h.p. at least once in the following number of iterations (Ghadimi and Lan, 2013):

$$\tilde{\mathcal{O}}\left( \frac{L(\mathcal{L}(\boldsymbol{\theta}_0) - \mathcal{L}^*)}{\epsilon^2} \left( 1 + \frac{\sigma^2}{b\epsilon^2} \right) \right), \tag{1}$$

where $\mathbb{E}_{i \in V}[(\mathbf{g}_{i,.} - \mathbf{g}_{V,.})^2] \leq \sigma^2$ is the variance of the individual gradients.

To improve the convergence of mini-batch SGD with mini-batch $b$, one can iteratively find weighted subsets (mini-batch coresets) of size $b$ that closely match the gradients of large random batches $\mathcal{M}_t^L$ of size $r > b$ (Yang et al., 2023). As the mini-batch coresets have a similar gradient to large batches, they have a smaller variance of $\sigma^2/r$. Thus, they improve the convergence of mini-batch SGD by $r/b$. The mini-batch coresets found by gradient matching are medoids (centroids) of the larger batch in the gradient space, weighted by their corresponding cluster size, and can be found by maximizing a monotone submodular[1] function via the greedy algorithm (Mirzasoleiman et al., 2020):

$$S_t^* \in \operatorname*{arg\,max}_{S \subset \mathcal{M}_t^L, |S| \leq b} \sum_{i \in \mathcal{M}_t^L} \max_{s \in S} [C - \|\mathbf{g}_{i,t} - \mathbf{g}_{s,t}\|], \tag{2}$$

where $C$ is a big constant. To find subsets efficiently, gradient of the loss w.r.t the input to the last layer of the model (which best captures the variation of gradient norm (Katharopoulos and Fleuret, 2018)) is commonly used (Mirzasoleiman et al., 2020; Pooladzandi et al., 2022; Yang et al., 2023).

## 4 CoLM: MINI-BATCH CORESETS FOR TRAINING LLMs

Despite its success of image classification tasks (Yang et al., 2023), the above gradient matching formulation performs poorly for training LLMs, due to the following reasons:

---

[1] A set function $F : 2^V \to \mathbb{R}^+$ is *submodular* if $F(e|S) = F(S \cup \{e\}) - F(S) \geq F(T \cup \{e\}) - F(T)$, for any $S \subseteq T \subseteq V$ and $e \in V \setminus T$. $F$ is monotone if for all $S \subset T$, $F(S) \leq F(T)$.

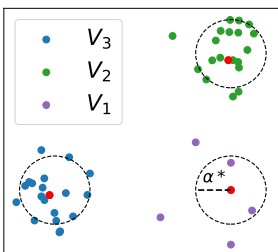 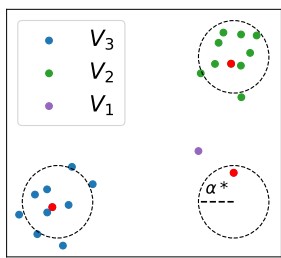 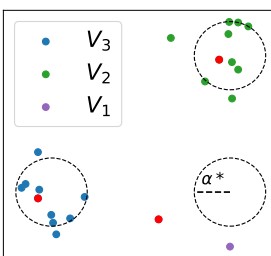

Figure 1: A toy imbalance data. (Left) Full data $V$ with two big (blue, green) and one small sources (purple). $k = 3$ medoids of the data are shown in red. (Middle & right) Two random samples of the data, with their corresponding $k = 3$ medoids. The $\alpha^\star$-neighborhoods of big sources are dense and thus medoids of random samples contain central examples of the big sources. However, the medoids of random sample do not necessarily contain central examples of the small source.

- **Highly Imbalanced Language Data.** Language data often contain highly imbalanced sources (e.g. categories or types of instructions). For example, the ratio of the largest to smallest source size in the MathInstruct data is 300. Here, subsets found by gradient matching do not contain representative examples (medoids) of the small sources and yield suboptimal performance.

- **Adam optimizer.** Adam (Kingma and Ba, 2014) is the commonly used optimizer for language tasks. Adam scales each gradient dimension to speed up learning along flatter dimensions and slow down learning along sharper ones. Matching the vanilla gradient of large batches yields suboptimal performance when training with Adam.

- **Very Large Gradient Dimensionality.** Finding medoids via Eq. (2) requires calculating pairwise distances between per-example gradients. But, in the very high-dimensional gradient space of LLMs, distances become vacuous. Even the last layer is too high-dimensional to find medoids accurately. For instance, the dimensionality of the last $V$-projection matrix of Phi-2 is 6.5M when training the full parameters and 327K (matrix B) when using LoRA with rank 128.

Next, we will discuss how we address each of the above challenges.

## 4.1 DEALING WITH THE IMBALANCED LANGUAGE DATA

First, we address the challenge of dealing with language data containing highly imbalanced mixture of sources. Our key observation is that larger random batches contain many examples from the big sources. Thus, mini-batch coresets selected via gradient matching contain central examples of the big source. In this case, training on the coresets benefits learning other examples in the big sources. On the other hand, small sources have a few examples in the large batches. Hence, the mini-batch coresets do not contain central examples of the small sources. In this case, training on the coresets does not benefit learning other examples from small sources. This implies that one cannot reliably select representative examples of small sources from the larger batches. Indeed, for optimal performance, it is crucial to train *on all examples of small sources* in the larger batches.

Next, we formalize this problem. Consider a dataset $V$ containing $Q$ sources, i.e., $V = \{V_1 \cup \cdots \cup V_Q\}$. Suppose gradients of examples in source $V_q$ at iteration $t$ are drawn from an underlying infinite set, according to an unknown probability distribution. Let $A_q^t$ be the set of $k$ medoids of the *infinite* set, such that around each $i \in A_q^t$ there is a neighborhood of radius at least $\alpha^*$, where the probability density is at least $\beta$ at all points, for some constants $\alpha^*, \beta$. This implies that the medoids are from reasonably dense and therefore representative regions of the gradient space. Let us consider $g : \mathbb{R} \to \mathbb{R}$, to be the volume of a ball of radius $\alpha$ centered at a point in the metric space. The following theorem shows that for a large enough source that is randomly partitioned into $m$ parts, there are many examples from the dense neighborhoods in every partition.

**Theorem 4.1.** *Let examples in $V_q$ be partitioned into $m$ parts. A number of examples $|V_q| \geq \frac{2km \log(km/\delta)}{\beta g(\alpha)}$, where $\alpha \leq \alpha^\star$, is suffice to (1) have at least $km \log(km/\delta)$ elements in the $\alpha$-neighborhood of each $i \in A_q^t$ and (2) have each partition contain elements from all $k$ $\alpha$-neighborhoods with probability at least $(1 - \delta)$ for a small $\delta > 0$.*

Next, we show that the medoids of every partition are central examples from the dense neighborhoods of the (infinite) data, with a high probability. That is, they are in $\alpha$-neighborhood of $A_q^t$.

**Theorem 4.2.** *Let $\delta, \epsilon > 0$ and let $n_q = |V_q|$ and $n_0$ be an integer such that for $n_q \geq n_0$ we have $\frac{n_q}{\ln(n_q)} \geq \frac{mk}{\epsilon^2}$. If $n_q \geq \max\left(n_0, \frac{m\log(2m/\delta)}{\epsilon^2}\right)$, with a probability of at least $1 - \delta$, medoids found by the greedy algorithm from every partition are in $\alpha$-neighborhoods of each $i \in A_q^t$.*

Note that in every large random batch, there is a part of every source $V_q$, with expected number of examples $|V_q|/|V|$. Hence, the above theorems imply that for small sources without a large-enough sample size, coresets found by gradient matching do not necessarily contain their central examples. Hence, training on them yields a poor performance on small sources. Fig 1 shows an illustration.

**Coresets for Imbalanced Data.** Consider a data $V = \{V_1 \cup \cdots \cup V_p \cup V_{p+1} \cup \cdots V_Q\}$, with $p$ small and $Q - p$ large sources. For a dataset with $c$ sources, we regard **small sources** as those with less than $|V|/c$ examples. To learn the small sources, we include all of their examples from the large batch in the small mini-batch coreset. That is, $S_s^t = \{v \in \mathcal{M}_t^L | v \in \cup_{i \in [p]} V_i\}$. But, for every big source $V_q$ where $q \in \{p+1, \cdots, Q\}$, we apply the greedy algorithm to its examples in the larger batch $V_q^t = \{v \in \mathcal{M}_t^L | v \in V_q\}$ and add its medoids $S_q^t \in V_q^t$ to the small mini-batch coreset, where $b_q = |S_q^t|$ is proportional to the number of examples from $V_q$ in $\mathcal{M}_t^L$, i.e., $b_q = (b - |S_s^t|).|V_q^t|/(|\mathcal{M}_t^L| - |S_s^t|)$. The mini-batch coreset at step $t$ is $S_t = \{S_s^t \cup S_{p+1}^t \cup \cdots \cup S_Q^t\}$. We note that sources are mostly separable based on their gradients. Thus, one subset can be found from all examples of big sources in the larger batch. However, selecting subsets separately yields a slightly better performance, as we show in our experiments. Finally, to ensure learning various big and small groups at a more uniform speed, we assign uniform weights to all the selected examples. For datasets such as MathInstruct, the sources are labeled in the training data. For datasets without specified sources, we cluster the hidden state of the model to find sources during fine-tuning.

The following theorem shows that the small mini-batch coresets have a smaller variance compared to random mini-batches of the same size. Therefore, they guarantee superior convergence rate.

**Theorem 4.3** (Variance reduction). *Let the number of outliers which do not belong to any $k$ dense areas be $\kappa$. Let $\alpha_u > \alpha^\star$ be the largest distance from an outlier to any centroids. Assume that all the selected samples $S_q^t$ belong to the dense areas. The variance of the mini-batch coresets of size $b$ is smaller than the variance of the random subset of size $b$ by up to $\frac{\kappa}{m}(\alpha_u - \alpha^\star)(2\alpha^\star + \frac{\kappa}{m}(\alpha_u - \alpha^\star))$.*

## 4.2 FINDING CORESETS FOR TRAINING WITH ADAM OPTIMIZER

Next, we address finding mini-batch coresets for training with Adam optimizer. Adam adapts the learning rate across dimensions by scaling the gradient updates by square roots of exponential moving averages of squared past gradients. In doing so, it reduces the learning rate across sharp dimensions and increases the learning rate across flatter dimensions to improve convergence. Formally,

$$\boldsymbol{m}_t = \frac{\beta_1 \boldsymbol{m}_{t-1} + (1 - \beta_1)\mathbf{g}_t}{1 - \beta_1^t}, \quad \boldsymbol{v}_t = \frac{\beta_2 \boldsymbol{v}_{t-1} + (1 - \beta_2)\mathbf{g}_t^2}{1 - \beta_2^t}, \quad \boldsymbol{\theta}_t = \boldsymbol{\theta}_{t-1} - \eta \frac{\boldsymbol{m}_t}{\epsilon + \sqrt{\boldsymbol{v}_t}}. \tag{3}$$

For selecting mini-batch coresets for training with Adam, matching the vanilla gradient is not enough. To do so, we normalize every gradient dimension by the exponential average of its historical values. Additionally, as we only select medoids of big sources, we calculate the historical terms $\boldsymbol{m}, \boldsymbol{v}$ only based on the big groups' gradients, which we denote by $\hat{\boldsymbol{m}}, \hat{\boldsymbol{v}}$. This allows a more precise selection of the subsets, as we also confirm in our experiments. Formally, we select the medoids of the normalized gradients of big sources, by solving the following submodular facility location function:

$$S_t^{q*} \in \underset{S \subset V_q^t, |S| \leq b_q}{\arg\max} \sum_{i \in V_q^t} \max_{s \in S}[C - \|\frac{\hat{\boldsymbol{m}}_{t,i}}{\epsilon + \sqrt{\hat{\boldsymbol{v}}_{i,t}}} - \frac{\hat{\boldsymbol{m}}_{t,s}}{\epsilon + \sqrt{\hat{\boldsymbol{v}}_{s,t}}}\|]. \tag{4}$$

The very high dimensional gradient of LLMs makes solving Eq. (4) prohibitively expensive. Besides, in such a high dimensional space, pair-wise distances become vacuous. Next, we discuss lowering the gradient dimensionality to find medoids of big sources more accurately.

### 4.3 FINDING LOWER-DIMENSIONAL GRADIENT ESTIMATES

The very high-dimensional gradients of LLMs are very noisy. To find smoother lower dimensional gradients in a memory efficient manner, we use zeroth-order methods to calculate the gradient of the last $V$-projection matrix, and then sparsify it to lower its dimensionality. The $V$-projections matrix allows finding higher-quality subsets, as we will confirm in our experiments. Notably, this approach stacks well with memory-efficient training methods such as LoRA, as we will confirm empirically.

Simultaneous Perturbation Stochastic Approximation (SPSA) (Spall, 1992) is a zeroth-order technique that estimates the gradient as:

$$\hat{\mathbf{g}} = \frac{\mathcal{L}(\boldsymbol{\theta} + \epsilon \boldsymbol{z}) - \mathcal{L}(\boldsymbol{\theta} - \epsilon \boldsymbol{z})}{2\epsilon} \boldsymbol{z} \approx \boldsymbol{z}\boldsymbol{z}^T \mathbf{g}, \tag{5}$$

where $\boldsymbol{z} \in \mathbb{R}^d$ is a random vector with $\boldsymbol{z} \sim \mathcal{N}(0, \boldsymbol{I}_d)$, and $d$ is the number of model parameters and $\epsilon$ is the perturbation scale. As $\epsilon \to 0$, the SPSA estimate provides a rank-1 reconstruction of the gradient, and this is smoother than the actual gradient calculated with backpropation. SPSA requires two forward passes through the model to compute the gradient estimate.

**Estimating the Last $V$-Projection Gradient.** To get the gradient of the last $V$-projection matrix for example $i$, instead of perturbing all the parameters, we sample random perturbations for parameters corresponding to the last (LoRA) $V$-projection in the perturbation vector $\boldsymbol{z}$, and use zero for the other entries:

$$\hat{\mathbf{g}}_{i,t}^{vp} = \frac{\mathcal{L}_i(\boldsymbol{\theta}_t + \epsilon \boldsymbol{z}_{vp}) - \mathcal{L}_i(\boldsymbol{\theta}_t - \epsilon \boldsymbol{z}_{vp})}{2\epsilon} \boldsymbol{z}_{vp}, \tag{6}$$

where $\boldsymbol{z}_{vp} \in \mathbb{R}^{d_{vp}}$ with $\boldsymbol{z} = [\boldsymbol{0}_{d-d_{vp}}, \boldsymbol{z}_{vp}]$ and $\boldsymbol{z}_{vp} \sim \mathcal{N}(0, \boldsymbol{I}_{d_{vp}})$, and $d_{vp}$ is the dimensionality of the flattened last $V$-projection matrix. Eq. (6) can be calculated very efficiently in just *one* forward pass. To do so, we first make a forward pass to get the activations $\boldsymbol{X}_{L-1}$ of the penultimate layer of the model. Then, we perturb the last-layer parameters twice to calculate $\hat{\mathbf{g}}_{i,t}^{vp}$ based on the pre-calculated $\boldsymbol{X}_{L-1}$. The time of getting the lower dimensional last-layer gradients will be dominated by the time of computing $\boldsymbol{X}_{L-1}$, and the cost of the second step is negligible. To minimize the memory requirement, one can follow (Malladi et al., 2023) to use a fix seed to generate the same perturbation $\boldsymbol{z}_{vp}$ multiple times. Hence, the memory overhead is also negligible.

We use the zeroth-order gradient estimates in Eq. (6) to calculate normalized gradients $\hat{\boldsymbol{m}}_t, \hat{\boldsymbol{v}}_t$ for the big sources. Nevertheless, these gradients are still too high-dimensional to find medoids accurately.

**Sparsifying the Last-layer Normalized Gradient Estimates for Adam.** To further reduce the gradient dimensionality, we sparsify the normalized $V$-projection gradients. The subsets are selected for each big source separately. Thus, for every big source $q$, we find dimensions that best preserve the normalized gradient norm of its examples $V_q^t \subseteq \mathcal{M}_t^L$ in the larger random batch. The normalized gradient norm to the first order indicates how much each gradient update achieves a loss reduction:

$$\Delta\mathcal{L}(\boldsymbol{\theta}) = \lim_{\epsilon \to 0} \frac{\mathcal{L}(\boldsymbol{\theta} + \epsilon \boldsymbol{m}/(\epsilon + \sqrt{\boldsymbol{v}_t})) - \mathcal{L}(\boldsymbol{\theta})}{\epsilon} = (\frac{\boldsymbol{m}}{\epsilon + \sqrt{\boldsymbol{v}}})^T \frac{\boldsymbol{m}}{\epsilon + \sqrt{\boldsymbol{v}}}. \tag{7}$$

Dimensions that best preserve the normalized gradient norm are those with the largest magnitude. Therefore, for every big source, we sparsify the normalized zeroth-order gradient $\hat{\boldsymbol{m}}_t/(\epsilon + \sqrt{\hat{\boldsymbol{v}}_t})$ of the last (LoRA) $V$-projection by a mask vector $M_q^t$, which has 1 for the top $h$ parameters with the largest magnitudes and 0 elsewhere.

**Using $\ell_1$ distance in high dimensions.** We calculate the pair-wise normalized gradient dissimilarity between sparsified gradients using $\ell_1$ distance, which is preferable to Euclidean distance in high dimensions (Aggarwal et al., 2001). The medoids for each big source are found by solving:

$$S_q^{t*} \in \underset{S \subset V_q^t, |S| \leq k_i}{\arg\max} \sum_{i \in V_q^t} \max_{s \in S} [C - \| \frac{\hat{\boldsymbol{m}}_{t,i}}{\epsilon + \sqrt{\hat{\boldsymbol{v}}_{t,i}}} \odot M_q^t - \frac{\hat{\boldsymbol{m}}_{t,s}}{\epsilon + \sqrt{\hat{\boldsymbol{v}}_{t,s}}} \odot M_q^t \|_1]. \tag{8}$$

In our experiments, we show that selecting as small as $0.7\%$ of the (LoRA) last layer gradient dimensions enables finding high-quality subsets efficiently. Additionally, we show that compared to random projection (Johnson, 1984), this approach enables finding higher-quality subsets and is orders of magnitude faster in practice.

The pseudo-code of CoLM is illustrated in Alg. 1 in Appendix B.

Table 1: Accuracies ($\uparrow$) on in-domain and out-of-domain datasets when fine-tuning Phi-2 with LoRA on the MathInstruct for 1K iterations. One-shot selection techniques (CL, GN, LC, FL, MP) are adapted to select small mini-batches on the fly. CoLM with batch size (bs = 64) outperforms all the baselines. Notably, CoLM even outperforms fine-tuning with bs = 256, while using 45% less memory, and achieves similar performance to fine-tuning for 2K iterations with bs = 128.

| Method | In-domain | | | Avg | SVAMP | Math. | SimulEq | Avg | Avg All |
|---|---|---|---|---|---|---|---|---|---|
| | GSM8K | MATH | NumGLUE | | | Out-domain | | | |
| Pretrained | 52.9 | 16.4 | 35.0 | 34.8 | 67.9 | 31.9 | 28.8 | 42.9 | 38.8 |
| FT (bs=64) | $66.5_{\pm0.8}$ | $28.4_{\pm0.3}$ | $50.2_{\pm0.9}$ | $48.3_{\pm0.2}$ | $79.2_{\pm0.4}$ | $52.4_{\pm0.8}$ | $24.1_{\pm1.5}$ | $51.9_{\pm0.2}$ | $50.1_{\pm0.2}$ |
| CL | $56.1_{\pm3.2}$ | $26.4_{\pm0.5}$ | $32.9_{\pm3.2}$ | $38.5_{\pm2.2}$ | $33.3_{\pm4.9}$ | $46.9_{\pm4.9}$ | $11.9_{\pm3.2}$ | $30.7_{\pm2.8}$ | $34.6_{\pm2.5}$ |
| BL | $58.0_{\pm0.5}$ | $21.1_{\pm0.5}$ | $43.7_{\pm2.0}$ | $40.9_{\pm0.8}$ | $77.1_{\pm0.7}$ | $38.0_{\pm4.4}$ | $18.4_{\pm0.6}$ | $44.5_{\pm1.3}$ | $42.7_{\pm1.0}$ |
| GN | $65.0_{\pm1.2}$ | $24.9_{\pm1.0}$ | $45.5_{\pm1.2}$ | $45.1_{\pm1.1}$ | $76.7_{\pm1.3}$ | $42.9_{\pm2.7}$ | $16.8_{\pm2.3}$ | $45.5_{\pm2.0}$ | $45.3_{\pm1.6}$ |
| LC | $59.3_{\pm0.9}$ | $24.0_{\pm0.7}$ | $48.0_{\pm0.5}$ | $43.8_{\pm0.4}$ | $79.5_{\pm0.8}$ | $45.9_{\pm0.3}$ | $23.6_{\pm2.6}$ | $49.7_{\pm1.1}$ | $46.7_{\pm0.7}$ |
| FL | $68.0_{\pm1.1}$ | $29.2_{\pm0.3}$ | $51.4_{\pm1.3}$ | $49.5_{\pm0.7}$ | $\mathbf{80.4_{\pm0.3}}$ | $55.6_{\pm1.2}$ | $30.5_{\pm2.5}$ | $55.5_{\pm1.3}$ | $52.5_{\pm1.0}$ |
| MP | $65.3_{\pm0.2}$ | $28.4_{\pm0.2}$ | $54.6_{\pm1.6}$ | $49.4_{\pm0.5}$ | $79.8_{\pm0.9}$ | $53.6_{\pm1.0}$ | $36.6_{\pm3.0}$ | $56.7_{\pm0.9}$ | $53.0_{\pm0.7}$ |
| **CoLM (Ours)** | $\mathbf{68.4_{\pm0.3}}$ | $\mathbf{29.8_{\pm0.4}}$ | $\mathbf{57.3_{\pm0.4}}$ | $\mathbf{51.9_{\pm0.3}}$ | $80.2_{\pm1.0}$ | $\mathbf{59.8_{\pm1.1}}$ | $\mathbf{44.1_{\pm2.8}}$ | $\mathbf{61.4_{\pm1.6}}$ | $\mathbf{56.6_{\pm0.9}}$ |
| FT (bs=128) | $67.4_{\pm0.5}$ | $28.8_{\pm0.3}$ | $53.2_{\pm1.2}$ | $49.8_{\pm0.5}$ | $80.4_{\pm1.3}$ | $55.6_{\pm0.4}$ | $29.9_{\pm2.4}$ | $55.3_{\pm1.0}$ | $52.6_{\pm0.6}$ |
| FT (bs=256) | $67.5_{\pm0.1}$ | $29.6_{\pm0.2}$ | $58.3_{\pm1.2}$ | $51.8_{\pm0.4}$ | $79.8_{\pm1.1}$ | $56.3_{\pm0.7}$ | $40.5_{\pm2.1}$ | $58.9_{\pm1.2}$ | $55.3_{\pm0.5}$ |
| FT (bs=128) 2K | $67.7_{\pm0.8}$ | $30.3_{\pm0.4}$ | $58.4_{\pm0.8}$ | $52.1_{\pm0.3}$ | $79.5_{\pm0.4}$ | $57.9_{\pm0.5}$ | $45.5_{\pm0.7}$ | $60.9_{\pm0.4}$ | $56.5_{\pm0.3}$ |

## 5 EXPERIMENTS

In this section, we evaluate the performance of CoLM, for fine-tuning LLMs, by comparing the performance, memory requirement, and wall-clock training time of training with small and large random mini-batches, with that of our method. In addition, we conduct an ablation study showing the effectiveness of different design choices of CoLM. We further demonstrate the effectiveness of CoLM in the pre-training setting in Appendix F.

### 5.1 SETTINGS

**Training datasets.** We use the MathInstruct (Yue et al., 2023) dataset which consists of about 260K instruction tuning examples, curated from 14 highly imbalanced open-source math datasets, with broad coverage of mathematical fields and a wide range of difficulty levels. The ratio of the largest to smallest source in MathInstruct is almost 300, and the distribution of sources can be found in Fig 4a in the Appendix. For datasets without specific sources, we use three datasets from the SuperGLUE benchmark (Wang et al., 2019) for the classification task: SST-2, CB, and MultiRC.

**Models.** We utilize the Phi-2, Phi-3 (Li et al., 2023b), Zephyr (Tunstall et al., 2023), and Llama-3 models (Dubey et al., 2024).

**Training details.** We use LoRA with a rank of 128, alpha of 512, and dropout rate of 0.05. For Phi models, we apply LoRA to all attention matrices (i.e. QKV_proj) and two fully connected layers while for Zephyr, we apply LoRA to all attention matrices (i.e. QKVO_proj). All experiments are run on 4 NVIDIA A40 GPUs. We repeat each experiment three times.

**Baselines.** We compare CoLM with normal fine-tuning (FT) using small and large batch sizes and an online selection method called Big Loss (BL) (Jiang et al., 2019). In addition, we adapt one-shot selection techniques, including Grad Norm (GN) (Katharopoulos and Fleuret, 2018), Middle Perplexity (MP) (Marion et al., 2023), Completion Length (CL) (Zhao et al., 2024a), Least Confidence (LC), and selecting centroids of the model's hidden state by maximizing submodular facility location (FL) (Bhatt et al., 2024), to select small mini-batches from larger ones during fine-tuning.

**Evaluation datasets.** Following (Yue et al., 2023), we use a variety of popular datasets across both in-domain and out-of-domain datasets. The in-domain datasets include GSM8K (Cobbe et al., 2021), MATH (Hendrycks et al., 2021), and NumGLUE (Mishra et al., 2022). For the out-of-domain datasets, we include SVAMP (Patel et al., 2021), Mathematics (Davies et al., 2021), and SimulEq (Koncel-Kedziorski et al., 2016).

Additional training details and evaluation metrics are specified in Appendix C.

### 5.2 MAIN RESULTS: MATHINSTRUCT

**CoLM achieves a superior performance.** Table 1 shows the in-distribution and out-of-distribution accuracies of fine-tuning Phi-2 with LoRA on the MathInstruct dataset for 1K iterations. First, we

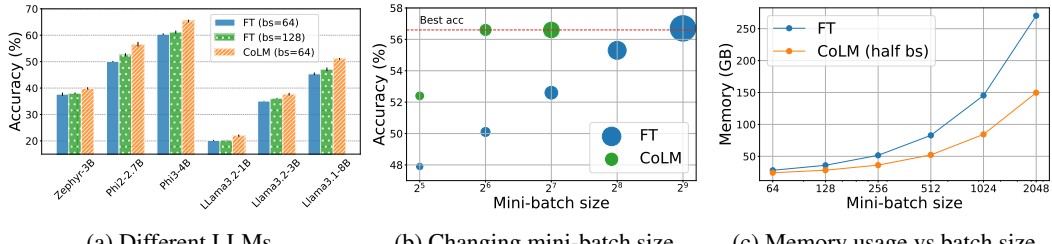

Figure 2: (a) CoLM with bs = 64 (from 128) outperforms fine-tuning different models with bs = 64 and bs = 128 by a large margin; (b) CoLM improves the performance of training with different batch sizes. The size of each circle is proportional to the training time of the corresponding method. (c) CoLM reduces memory consumption, with reduction increasing as the batch size grows.

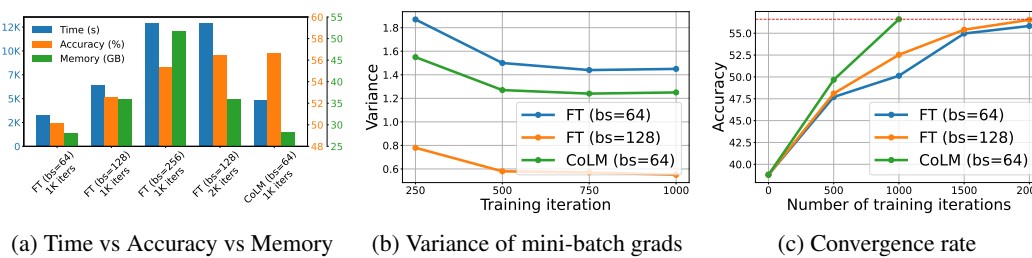

Figure 3: Fine-tuning Phi-2 on MathInstruct. (a) Wall-clock time (including the time for CoLM's selection), memory consumption, and performance of fine-tuning. CoLM outperforms normal fine-tuning for 1K iterations with bs = 128 (256), while being 1.3x (2.7x) faster and consuming 20% (45%) less memory, respectively; (b) CoLM has a smaller variance than random mini-batches of the same size; (c) CoLM converges much faster than normal fine-tuning (FT).

see that using a larger mini-batch size improves the performance, which is consistent with the theoretical results in Eq. (1). Besides, we note that training for 1K steps with a mini-batch size of 64, 128, 256 corresponds to training on 25%, 50% and 100% of the data, respectively. Remarkably, training on only 25% of the data with CoLM with bs = 64 (selected from 128) outperforms all the baselines and achieves a similar performance to fine-tuning for 2K iterations with bs = 128. Interestingly, training for 1K iterations with CoLM using bs = 64 even outperforms training with bs = 256.

**CoLM improves the performance of different models and batch sizes.** Figure 2a shows that CoLM significantly outperforms normal fine-tuning across 6 different model architectures. Specifically, fine-tuning Phi-3 for 4K iterations using CoLM with bs = 64 outperforms normal fine-tuning for 4K iterations with bs = 64 and bs = 128 by 5% and 4.2%. Notably, for better-performing models, CoLM provides more performance improvement. This confirms its applicability to state-of-the-art architectures. Fig. 2b shows that CoLM improves the performance of different batch sizes, including bs = 32, bs = 64, and bs = 128, without significantly increasing the training time.

**CoLM effectively reduces the Activation Memory.** The memory required for training an LLM can be decomposed into three parts: activation memory + weight memory + optimizer state memory. Memory efficient methods often reduce the weight or optimizer-state memory. For example, LoRA reduces the optimizer state memory but slightly increases the weight and activation memory by adding low-rank matrices. Orthogonal to such methods, CoLM effectively reduces the activation memory by reducing the batch size. Hence, stacked with memory-efficient methods such as LoRA and gradient accumulation, it can further reduce the memory, particularly when batch size is large. In that scenario, the activation memory dominates the optimizer state and weight memory. Figure 2c shows the memory usage of normal fine-tuning and CoLM with half the batch size. Notably, for total bs = 2048, CoLM (bs = 1024) requires almost 2x less memory than fine-tuning with bs=2048. A larger batch size is useful in particular for pre-training. Our experiments in Appendix F, confirm the benefits of CoLM to pre-training. Furthermore, while memory efficient methods harm the performance, CoLM effectively improves the performance over training with *larger batches*. A detailed analysis of memory overhead of our method can be found in Appendix E.

**CoLM speeds up training and improves convergence.** Fig. 3a compares the wall-clock time and average performance of CoLM and normal fine-tuning Phi-2, using LoRA. For CoLM, the

Table 2: Effect of different components in CoLM.

| Method | In-domain | Out-domain | Avg |
|---|---|---|---|
| Weighted medoids | $48.5_{\pm1.1}$ | $53.8_{\pm0.7}$ | $51.1_{\pm0.9}$ |
| Medoids (using cosine distance) | $48.7_{\pm0.3}$ | $54.0_{\pm1.6}$ | $51.3_{\pm0.9}$ |
| Medoids (using $\ell_1$ distance) | $48.6_{\pm0.3}$ | $54.4_{\pm0.8}$ | $51.5_{\pm0.5}$ |
| Medoids of big sources & keep small sources | $50.9_{\pm1.0}$ | $58.4_{\pm0.8}$ | $54.6_{\pm0.6}$ |
| Medoids of big sources selected separately & keep small sources | $50.6_{\pm0.2}$ | $59.6_{\pm0.9}$ | $55.1_{\pm0.5}$ |
| **CoLM**: Medoids of big sources selected separately for Adam & keep small sources | $\mathbf{51.9_{\pm0.3}}$ | $\mathbf{61.4_{\pm1.6}}$ | $\mathbf{56.6_{\pm0.9}}$ |

wall-clock time includes the time for selecting the mini-batches. Remarkably, CoLM with bs = 64 (selected from 128) speeds up training for 2K iterations with bs = 128 and 1K iterations with bs = 256 by 2.7x, while having 20% and 45% less memory requirements and superior performance. Figure 3b shows that throughout training, the variance of CoLM (bs = 64) gradients is smaller than normal fine-tuning with bs = 64, which confirms our theoretical results in Sec. 4.1. This yields a faster convergence compared to random mini-batches of the same size, as shown in Fig 3c. At the same time, although random bs = 128 has a lower variance than CoLM with bs = 64, the more uniform speed of learning sources by CoLM enables it to obtain a superior performance. Furthermore, we show that CoLM yields smaller loss compared to baselines throughout training and achieves the optimal performance in less training time in Appendix D.

## 5.3 ABLATION STUDIES

**The importance of different components.** Tab 2 highlights the importance of different components in CoLM. Notably, including all examples of the small data sources improves the accuracy significantly by around 3% on average. This finding well aligns with our analysis in Sec. 4.1. Selecting subsets separately per source and normalizing gradients for Adam further boost the performance by 0.5% and 1.5%, respectively, justifying our methods in Sections 4.2 and 4.3. Using uniform weights for selected example slightly improves the performance by 0.4%.

**Sparsification criteria for $V$-projection.** Table 3 compares the performance of CoLM for different sparsification criteria. Keeping parameters with the largest gradient magnitude achieves the best performance. This result is consistent with that of (Guo et al., 2024), which show that parameters with the largest gradient magnitude are the most salient. Weight magnitude, a common approach in network pruning (Han et al., 2015), yields a slightly lower performance than random sparsification.

**Sparsity level.** Table 4 illustrates the performance of CoLM when changing the dimensionality ($h$) of the sparsified gradients. The accuracy peaked at $h = 2560$, which equals the dimensionality of the hidden state of Phi-2, and gradually decreases for larger values of $h$. This result is expected as gradients in high dimension suffer from the curse of dimensionality, yielding a sub-optimal solution.

**Choices of low-dimensional gradient approximations.** We compare the usage of sparsified MeZO gradient in our method with the sparsified actual gradient (via backprop), projected actual gradient, and low-rank MeZO gradient. For sparsified actual gradient, we apply the same sparsification technique as our CoLM. For projected actual gradient, we apply a random projection to the actual gradient and leverage the memory-efficient implementation introduced by Park et al. (2023). For low-rank MeZO gradient, we use the low-rank projection technique with SVD in GaLore Zhao et al. (2024b) and adopt their best setting with rank $r = 8$ and subspace change frequency $T = 200$. Table 5 shows that the sparsified MeZO gradient has a clear margin over the other choices of low-dimensional gradient estimates, highlight the effectiveness of zero-th order gradient and our sparsification strategy in selecting high-quality subsets.

**Choices of layers.** We replace the last $V$-projection layer with the last FC layer and the combination of the last $Q, K, V$ projections in CoLM. As can be seen in Table 6, using the MeZO gradient of the last $V$ projection matrix yields a gap of almost 2% compared to other choices of layers.

**Completion length.** (Zhao et al., 2024a) found that fine-tuning on examples with the longest completion length improves the performance. Figure 4b in the Appendix shows that CoLM, in contrast, selects examples with shorter answers (avg length $\approx 120$) than the average completion length of the data, which is about 130. Nevertheless, CoLM significantly improves over selecting examples with the longest completion length (avg length $\approx 210$) in random mini-batch as indicated in Table 1.

Table 3: Effect of the sparsification criteria.

| Criteria | In-domain | Out-domain | Avg |
|---|---|---|---|
| random | $51.1_{\pm 0.9}$ | $59.6_{\pm 2.5}$ | $55.4_{\pm 1.7}$ |
| weight | $51.1_{\pm 0.7}$ | $59.6_{\pm 0.1}$ | $55.3_{\pm 0.3}$ |
| weight $\times$ grad | $51.4_{\pm 0.3}$ | $58.8_{\pm 1.5}$ | $55.1_{\pm 0.9}$ |
| grad | $\mathbf{51.9_{\pm 0.3}}$ | $\mathbf{61.4_{\pm 1.6}}$ | $\mathbf{56.6_{\pm 0.9}}$ |

Table 4: Effect of the sparsity level.

| Dim | In-domain | Out-domain | Avg |
|---|---|---|---|
| 1280 | $50.4_{\pm 0.8}$ | $57.8_{\pm 1.8}$ | $54.1_{\pm 1.3}$ |
| 2560 | $\mathbf{51.9_{\pm 0.3}}$ | $\mathbf{61.4_{\pm 1.6}}$ | $\mathbf{56.6_{\pm 0.9}}$ |
| 5120 | $51.2_{\pm 0.1}$ | $60.1_{\pm 0.9}$ | $55.7_{\pm 0.5}$ |
| 10240 | $51.6_{\pm 0.5}$ | $59.4_{\pm 0.7}$ | $55.5_{\pm 0.4}$ |

Table 5: Comparison between different low-dimensional gradient approximations.

| Approx | In-domain | Out-domain | Avg |
|---|---|---|---|
| Sparsified actual grad | $51.0_{\pm 0.3}$ | $58.3_{\pm 0.3}$ | $54.7_{\pm 0.3}$ |
| Projected actual grad | $50.9_{\pm 0.5}$ | $59.4_{\pm 0.4}$ | $55.2_{\pm 0.4}$ |
| Low-rank MeZO grad | $51.0_{\pm 0.2}$ | $58.1_{\pm 0.8}$ | $54.6_{\pm 0.4}$ |
| Sparsified MeZO grad | $\mathbf{51.9_{\pm 0.3}}$ | $\mathbf{61.4_{\pm 1.6}}$ | $\mathbf{56.6_{\pm 0.9}}$ |

Table 6: Comparison between different choices of layers.

| Layer(s) | In-domain | Out-domain | Avg |
|---|---|---|---|
| FC | $50.6_{\pm 0.8}$ | $58.4_{\pm 0.6}$ | $54.5_{\pm 0.1}$ |
| V proj | $\mathbf{51.9_{\pm 0.3}}$ | $\mathbf{61.4_{\pm 1.6}}$ | $\mathbf{56.6_{\pm 0.9}}$ |
| QKV projs | $51.3_{\pm 1.0}$ | $58.2_{\pm 1.1}$ | $54.7_{\pm 1.0}$ |

Table 7: Accuracies ($\uparrow$) when fine-tuning Phi-2 with LoRA on three datasets from the SuperGLUE benchmark for 80 iterations. CoLM with bs = 64 (from 128) effectively improves the performance of normal fine-tuning with bs = 64. When using clusters found by the fine-tuned model, CoLM outperforms fine-tuning with bs = 128.

| | SST-2 | CB | MultiRC | Avg |
|---|---|---|---|---|
| Pretrained | 56.6 | 45.5 | 46.3 | 49.5 |
| FT (bs=64) | $91.4_{\pm 0.2}$ | $69.1_{\pm 1.8}$ | $62.0_{\pm 2.2}$ | $74.2_{\pm 1.4}$ |
| **CoLM** (clustering during fine-tuning) | $92.3_{\pm 0.4}$ | $73.3_{\pm 1.7}$ | $70.5_{\pm 3.6}$ | $78.7_{\pm 1.9}$ |
| **CoLM** (clustering of the fine-tuned model) | $\mathbf{92.4_{\pm 0.7}}$ | $\mathbf{77.6_{\pm 3.7}}$ | $\mathbf{73.0_{\pm 3.4}}$ | $\mathbf{81.0_{\pm 2.6}}$ |
| FT (bs=128) | $92.1_{\pm 1.0}$ | $72.1_{\pm 0.8}$ | $72.6_{\pm 5.2}$ | $78.9_{\pm 2.3}$ |

## 5.4 DATASETS WITHOUT SPECIFIC SOURCES: SUPERGLUE BENCHMARK

We apply CoLM to fine-tuning Phi-2 with bs = 64 (selected from 128) for 80 iterations on three classification datasets (SST-2, CB, MultiRC) in the SuperGLUE benchmark. Note that the Super-GLUE datasets do not have any source information. To find sources, we warm up the model for 20 iterations with bs = 64, and then cluster the model's hidden states. We consider each cluster as a source and define small sources as those with less than $|V|/c$ examples, where $c$ is the number of clusters. We update the clustering four times during fine-tuning. As shown in Table 7, CoLM outperforms normal fine-tuning with bs = 64 by 4.5% on average. This shows CoLM's applicability to datasets without specified sources. Additionally, we find that clusters found by the fine-tuned model can significantly enhance the results. Compared to updating the clusters during training, using the clusters by a fine-tuned model improved the performance by 6.8% on average. Compared to standard fine-tuning with bs = 128, CoLM also improved the performance by 2.1%. To leverage this, one can fine-tune a smaller proxy model on a smaller random subset of the data and cluster its hidden states to find sources more accurately, without a large overhead. For SST-2 and MultiRC, we trained the model on a randomly selected subset of 3,000 examples to find the clusters.

## 6 CONCLUSION

To simulate training with larger mini-batch sizes with limited memory, an effective approach is to find small mini-batch coresets that match the gradient of larger random batches. We showed that for language data with highly imbalanced sources, mini-batch coresets found by gradient matching do not contain representative examples of the small sources. Thus, one should keep all examples of the small sources and augment them with examples that match the gradient of big sources in the larger batch. To enable solving the gradient matching problem effectively, we used techniques from zeroth-order optimization and model pruning to find lower-dimensional gradient estimates. We also showed that matching the normalized gradient of larger batches provides superior performance for training with Adam. Our method, CoLM, outperforms fine-tuning Phi models on MathInstruct with 4x larger batch size, while being 2.7x faster, and also improves fine-tuning on the SuperGLUE benchmark.

## ACKNOWLEDGMENTS

This research was partially supported by the National Science Foundation CAREER Award 2146492, the NSF-Simons AI Institute for Cosmic Origins, and an Okawa Research Award.

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

# A   VARIANCE REDUCTION BY FACILITY LOCATION

**Outline.** In this section, we present the proofs for our theoretical results in Section 4.1. Firstly, we introduce the notations, problem formulation, and all assumptions. Secondly, Theorem 4.1 is the result of Lemma A.4 and the first part of Lemma A.6. In addition, we provide a bound for the local optimal solution in Corollary A.7. Thirdly, we prove Theorem 4.2 using Lemma A.8. Finally, Theorem 4.3 is subsequent to Lemma A.9.

**Notations.** $d(\cdot, \cdot) : V \times V \to \mathbb{R}$ is the distance between two elements. $N_\alpha(v) = \{w : d(v, w) \leq \alpha\}$ is the set of elements within a distance $\alpha$ from $v$, called $\alpha$-neighborhood. $g(\alpha) : \mathbb{R} \to \mathbb{R}$ is the volume of a ball radius $\alpha$ centered at a point in the metric space. In $\mathbb{R}^D$, we have $g(\alpha) = O(\alpha^D)$.

**Facility Location.** For a set $V$, we solve the k-medoid problem by finding a subset $S$ such that $|S| = k$ and $S$ minimizes $L(S) = \frac{1}{|V|} \sum_{v \in V} \min_{e \in S} d(v, e)$. We can turn $L$ into a monotone submodular function by using an auxiliary element $v_0$: $f(S) = L(\{v_0\}) - L(S \cup \{v_0\})$.

**Settings.** We have a dataset $V$ with $n$ examples in $\mathbb{R}^D$ which is drawn from an underlying infinite set, according to some unknown probability distribution. Let $A$ such that $|A| = k$ be the global optimal solution of the facility location problem in the infinite set.

**Assumption A.1** (Data structure). For each $e_i \in A$, there is a neighborhood of radius at least $\alpha^\star$, where the probability is at least $\beta$ at all points, for some constant $\alpha^\star$ and $\beta$.

It is known that $f$ is decomposable. In other words, $f$ can be written as sum of (non-negative) monotone submodular functions as follows: $f(S) = \frac{1}{|V|} \sum_{v \in V} f_v(S)$. We define the evaluation of $f$ restricted to $D \subseteq V$ as follows: $f_D(S) = \frac{1}{|D|} \sum_{i \in D} f_i(S)$. Assume that the objective function $f$ have the following two properties.

**Assumption A.2** (Lipschitz property). $f : 2^V \to \mathbb{R}$ is $\lambda-$Lipschitz. In other words, for equal sized sets $S = \{v_1, v_2, \ldots, v_k\}$ and $S' = \{v'_1, v'_2, \ldots, v'_k\}$ and for any matching of elements $M = \{(v_1, v'_1), (v_2, v'_2), \ldots, (v_k, v'_k)\}$, the difference between $f(S)$ and $f(S')$ is bounded by the total of distances between respective elements $|f(S) - f(S')| \leq \lambda \sum_i d(v_i, v'_i)$.

**Assumption A.3** (Bound property). $f_i$ is bounded, and without loss of generality $0 \leq f_i(S) \leq 1$ for $1 \leq i \leq |V|, S \subseteq V$.

The dataset is randomly partition into $m$ large mini-batches $\{\mathcal{M}_j\}_{j=1}^m$ of size $r = \frac{n}{m}$. We denote the local optimal solution for each $\mathcal{M}_j$ as $A_j$ where $|A_j| = k$. We are showing in the next Lemma that when the size of the training set is large enough, it has many examples from all the dense areas.

**Lemma A.4.** *A number of elements $n \geq \frac{2km \log(km/\delta)}{\beta g(\alpha)}$, where $\alpha \leq \alpha^\star$ suffices to have at least $km \log(km/\delta)$ elements in the $\alpha-$neighborhood of each $e_i \in A$ with probability at least (1 - δ), for small values of δ.*

*Proof.* The probability of a random element being in $N_\alpha(e_i)$ is at least $\beta g(\alpha)$. Thus, the expected number of $\alpha-$neighbors of an $e_i \in A$ is $E[|N_\alpha(e_i)|] \geq 2km \log(km/\delta)$.

From the Chernoff bound, we have for every $t < 0$,

$$
\begin{aligned}
P[|N_\alpha(e_i)| \leq km \log(km/\delta)] &\leq E[\exp(t * |N_\alpha(e_i)|)] \exp(-t * km \log(km/\delta)) \\
&\leq \exp(t * (E[|N_\alpha(e_i)|] - km \log(km/\delta))) \\
&\leq \exp(t * km \log(km/\delta)).
\end{aligned} \tag{9}
$$

Let $t = -\frac{1}{km}$ in the above equation, we have

$$
P[|N_\alpha(e_i)| \leq km \log(km/\delta)] \leq \exp(-\log(km/\delta)) = \frac{\delta}{km}. \tag{10}
$$

Therefore, the probability that some $e_i \in A$ does not have a large enough neighborhood is

$$
P[\bigcup_{i=1}^k |N_\alpha(e_i)| \leq km \log(km/\delta)] \leq \sum_{i=1}^k P[|N_\alpha(e_i)| \leq km \log(km/\delta)]
$$

$$
\leq k \frac{\delta}{km} = \frac{\delta}{m} \leq \delta. \tag{11}
$$

Therefore, with probability at least $1 - \delta$, the $\alpha-$neighborhood of each element $e_i \in A$ contains at least $km \log(km/\delta)$ elements. $\qquad\square$

Next, we prove that sampling with replacement guarantees that each mini-batch has elements from all the dense areas.

**Lemma A.5** (Sampling with replacement). *If for each $e_i \in A$, $|N_\alpha(e_i)| = m \log(k/\delta)$, and if $\mathcal{M}_j$ is a mini-batch of size $n/m$ sampling with replacement, then $\mathcal{M}_j$ contains elements from all $k$ dense areas with probability at least $(1 - \delta)$.*

*Proof.* The number of mini-batches $\mathcal{M}_j$ does not contain elements from $N_\alpha(e_i)$ is $(n - m \log(k/\delta))^{(n/m)}$. The total number of mini-batches of size $n/m$ is $n^{(n/m)}$. Thus, the probability of $\mathcal{M}_j$ does not contain elements from $N_\alpha(e_i)$ is $(\frac{n - m \log(k/\delta)}{n})^{(n/m)} \approx (1 - \frac{1}{\frac{n}{m \log(k/\delta)}})^{(n/m)} = exp(-\log(k/\delta)) = \delta/k$. Therefore, the probability that $\mathcal{M}_j$ does not contain elements from all $k$ dense areas is

$$P[\bigcup_{i=1}^{k} |\mathcal{M}_j \cap N_\alpha(e_i)| = 0] \leq \sum_{i=1}^{k} P[|\mathcal{M}_j \cap N_\alpha(e_i)| = 0] = \delta. \qquad (12)$$

$\qquad\square$

The above guarantee also holds for sampling without replacement as shown in the following lemma.

**Lemma A.6** (Sampling without replacement). *If for each $e_i \in A$, $|N_\alpha(e_i)| \geq km \log(km/\delta)$, and if $V$ is partitioned into $m$ mini-batch $\mathcal{M}_1, \mathcal{M}_2, \ldots, \mathcal{M}_m$, then each $\mathcal{M}_j$ contains elements from all the dense areas and $|f(A) - f(A_j)| \leq \lambda \alpha k$ with probability at least $(1 - \delta)$.*

*Proof.* Because $|N_\alpha(e_i)| \geq km \log(km/\delta)$, we can construct $k$ mutually disjoint subsets $\{S_i\}_{i=1}^{k}$ such that $S_i \in N_\alpha(e_i)$ and $|S_i| = m \log(km/\delta)$. Each element in $S_i$ goes into a particular $\mathcal{M}_j$ with a probability of $1/m$. The probability that a particular $\mathcal{M}_j$ does not contain an element in $S_i$ is $P[|\mathcal{M}_j \cap S_i| = 0] = (1 - 1/m)^{m \log(km/\delta)} = \frac{\delta}{km}$. The last equality hold because $\lim_{m \to +\infty} (1 - 1/m)^m = \exp(-1)$. The probability that $\mathcal{M}_j$ does not intersect with at least one $S_i$ is

$$P[\bigcup_{i=1}^{k} |\mathcal{M}_j \cap S_i| = 0] \leq \sum_{i=1}^{k} P[|\mathcal{M}_j \cap S_i| = 0] = \frac{\delta}{m}. \qquad (13)$$

Therefore, the probability that $\mathcal{M}_j$ contains elements from every $S_i$ is at least $1 - \frac{\delta}{m}$. Thus, the probability that every $\mathcal{M}_j$ contains elements from every $S_i$ is

$$P[\bigcap_{j=1}^{m} (\bigcap_{i=1}^{k} |\mathcal{M}_j \cap S_i| > 0)] = \prod_{j=1}^{m} P[\bigcap_{i=1}^{k} |\mathcal{M}_j \cap S_i| > 0] = (1 - \frac{\delta}{m})^m \approx 1 - \delta. \qquad (14)$$

Thus, with high probability $1 - \delta$, every $\mathcal{M}_j$ has a subset $S_j$ such that are $|S_j| = |A| = k$ and $|S_j \cap N_\alpha(e_i)| > 0$ for $e_i \in A$. Therefore, $f(A) - f(A_j) \leq f(A) - f(S_j) \leq \lambda \alpha k$. $\qquad\square$

From Lemmas A.4 and A.6, we have the following corollary

**Corollary A.7** (Bound for local optimal solution). *For $n \geq \frac{2km \log(4km/\delta)}{\beta g(\frac{\epsilon}{\lambda k})}$, where $\frac{\epsilon}{\lambda k} \leq \alpha^\star$, if $V$ is partitioned into $m$ mini-batches $\mathcal{M}_1, \mathcal{M}_2, \ldots, \mathcal{M}_m$, then for sufficiently small values of $\delta$, we have $|f(A) - f(A_j)| < \epsilon$ with a probability of at least $1 - \delta$.*

**Lemma A.8** (Bound for local evaluation). *Let $n_0$ be an integer such that for $n \geq n_0$ we have $\frac{n}{ln(n)} \geq \frac{mk}{\epsilon^2}$. If $n \geq \max \left(n_0, \frac{m \log(2m/\delta)}{\epsilon^2}\right)$, with a probability of at least $1 - \delta$, we can evaluate $f$ on each mini-batch $\mathcal{M}_j$ with a small error of $\epsilon$, i.e., $|f_{\mathcal{M}_j}(S) - f(S)| < \epsilon$.*

*Proof.* Note that each mini-batch has exactly $|\mathcal{M}_j| = n/m$ elements. Let us define $\xi_j(S)$ the event that $|f_{\mathcal{M}_j}(S) - f(S)| < \epsilon$, for some fixed $\epsilon < 1$ and a fixed $S$ with $|S| \leq k$. Note that $\xi_j(S)$ denotes the event that the empirical mean $f_{\mathcal{M}_j}(S)$ is close to the true mean. Because $f$ is decomposable, we have $f_{\mathcal{M}_j}(S) = \frac{1}{|\mathcal{M}_j|} \sum_{i \in \mathcal{M}_j} f_j(S) = \sum_{i \in \mathcal{M}_j} \frac{f_j(S)}{|\mathcal{M}_j|}$. Also remember that $0 \leq \frac{f_j(S)}{|\mathcal{M}_j|} \leq \frac{1}{|\mathcal{M}_j|}$. Based on the Hoeffding inequality (without replacement) we have

$$P[\neg \xi_i(S)] = P[f_{V_i}(S) - f(S) \geq \epsilon] = P[f_{V_i}(S) - E[f_{V_i}(S)] \geq \epsilon]$$
$$\leq 2 \exp\left(-\frac{2\epsilon^2}{|V_i|(\frac{1}{|V_i|} - 0)^2}\right)$$
$$= 2 \exp(-2\epsilon^2 |V_i|)$$
$$= 2 \exp(-2n\epsilon^2/m). \tag{15}$$

Let $\xi_i$ be an event that $|f_{V_i}(S) - f(S)| < \epsilon$ for any $S$ such that $|S| \leq k$. Note that there are at most $n^k$ sets of size at most $k$ (because sampling k samples with replacement results in a subset of size at most $k$). Hence,

$$P[\neg \xi_i] \leq 2n^k \exp(-2n\epsilon^2/m) \tag{16}$$

There are $m$ mini-batches, by the union bound we can conclude that

$$P[\bigcup_{i=1}^{m} \neg \xi_i] \leq \sum_{i=1}^{m} P[\neg \xi_i] \leq 2mn^k \exp(-2n\epsilon^2/m) \tag{17}$$

The above calculation implies that we need to choose $\delta \geq 2mn^k \exp(-2n\epsilon^2/m)$ so that w.h.p $1 - \delta$ we can evaluate $f$ locally on each mini-batch. For large $n$, the function $\frac{n}{ln(n)}$ is an increasing function, thus, there exists $n_0$ such that for $n \geq n_0$, $\frac{n}{ln(n)} \geq \frac{mk}{\epsilon^2}$. Then, we choose $n$ as follows

$$n = \max\left(n_0, \frac{m \log(2m/\delta)}{\epsilon^2}\right) \tag{18}$$

$\square$

For $|f_{V_i}(S) - f(S)| < \epsilon$ to hold for all subsets $S$ such that $|S| \leq k$, the data distribution of $V_i$ should be similar to that of $V$. Hence, $k$-medoids of $V_i$ are in close neighborhood of $k$-medoids of $V$.

**Lemma A.9** (Upper bound for the variance). *Let the number of outliers which do not belong to any $k$ dense area be $\kappa$. Let $\alpha_u > \alpha^\star$ be the largest distance from an outlier to any centroids. Assume that all the selected samples $A_j$ belong to the dense areas. The upper bound of the variance of the local optimal solution $A_j$ is smaller than that of the random subset of size $k$.*

*Proof.* For each subset $S$, we use the notation $S^c$ to denote the centroid of this subset. Let $\epsilon_j$ be the distance between the centroid of a subset $S_j$ of size $k$ to the centroid of $A$. The variance of the subset $S_j$ has an upper bound as follow.

$$\text{Var}(S_j^c) = \text{Var}(A^c + \epsilon_j)$$
$$= \text{Var}(\epsilon_j)$$
$$\leq E[\epsilon_j^2]. \tag{19}$$

For each local optimal solution $A_j$, we know that $\epsilon_j \leq \alpha^\star$. For a random subset $S_j$ of size $k$, there is $\kappa/m$ outliers and $k - (\kappa/m)$ examples from the dense areas in the subset on average. Thus, the distance $\epsilon_j$ is bounded as $\epsilon_j \leq (1 - \frac{\kappa}{m})\alpha^\star + \frac{\kappa}{m}\alpha_u \geq \alpha^\star$. Therefore, the upper bound of a random subset $S_j$ is larger than that of the select subset $A_j$. $\square$

From the above lemma, we can conclude that

**Theorem A.10** (Variance reduction). *The variance of the mini-batch coresets of size $b$ is smaller than the variance of the random subset of size $b$ by up to $\frac{\kappa}{m}(\alpha_u - \alpha^\star)(2\alpha^\star + \frac{\kappa}{m}(\alpha_u - \alpha^\star))$.*

---

**Algorithm 1** Coresets for Training LLMs (CoLM) on Imbalanced Language Data

---

1: **Input:** $\boldsymbol{\theta} \in \mathbb{R}^d$, loss $\mathcal{L} : \mathbb{R}^d \to \mathbb{R}$, step budget $T$, batch size $b$, learning rate schedule $\{\eta_t\}$, small sources $\{V_1, \cdots, V_p\}$, large sources $\{V_{p+1}, \cdots, V_Q\}$

2: **for** $t = 1, \cdots, T$ **do**

3:     Sample batch $\mathcal{M}_t \subset \mathcal{D}$

4:     $S_s^t \leftarrow \{v \in \mathcal{M}_t | v \in \bigcup_{i \in [p]} V_p\}$            // Keep all samples from small sources in the batch

5:     **for** $i \in \{p+1, \cdots, q\}$ **do**

6:         $V_i^t \leftarrow \{v \in \mathcal{M}_t | v \in V_i\}$           // Find all samples from each big source

7:         $b_i \leftarrow (b - |S_t^s|) \cdot |V_i^t| / (|\mathcal{M}_t| - |S_t^s|)$     // Calculate the number of selected samples

8:         Get the zeroth-order gradient $\hat{g}_{i,t}^{vp}$ of the last (LoRA) V-projection using Eq 6.

9:         Calculate the normalized gradient $\frac{\hat{\boldsymbol{m}}_{t,i}}{\epsilon + \sqrt{\hat{\boldsymbol{v}}_{t,i}}}$ from historical terms and zeroth-order gradient $\hat{g}_{i,t}^{vp}$ uisng Eq 3.

10:         Create a mask vector $M_q^t$ for top $h$ parameters with largest magnitude.

11:         $S_i^t \leftarrow \arg\max_{S \subset V_i^t, |S| \leq b_i} \sum_{i \in V_i^t} \max_{s \in S} [C - \|\frac{\hat{\boldsymbol{m}}_{t,i}}{\epsilon + \sqrt{\hat{\boldsymbol{v}}_{t,i}}} \odot M_i^t - \frac{\hat{\boldsymbol{m}}_{t,s}}{\epsilon + \sqrt{\hat{\boldsymbol{v}}_{t,s}}} \odot M_i^t\|_1]$.
    // Solve the submodular facility location optimization problem

12:     **end for**

13:     $S_t \leftarrow \{S_s^t \cup S_{p+1}^t \cup \cdots \cup S_q^t\}$

14:     $\boldsymbol{\theta} \leftarrow \boldsymbol{\theta} - \eta \nabla \mathcal{L}_{S_t}(\boldsymbol{\theta})$

15: **end for**

---

## B    Pseudo-code

Algorithm 1 illustrates the pseudo-code of our CoLM.

## C    Fine-tuning settings

**Training datasets.** We use the MathInstruct (Yue et al., 2023) dataset for the challenging task of mathematical reasoning. MathInstruct consists of about 260K instruction tuning examples, curated from 14 highly imbalanced open-source math datasets, with broad coverage of mathematical fields and a wide range of difficulty levels. The ratio of the largest to smallest source in MathInstruct is almost 300, and the distribution of sources can be found in Fig 4a in the Appendix. Fine-tuning on MathInstruct has shown state-of-the-art performance on a variety of standard math evaluation benchmarks. For datasets without specific sources, we use three datasets from the SuperGLUE benchmark (Wang et al., 2019) for the classification task: SST-2, CB, and MultiRC. For CB, we use the full training dataset, which consists of 250 examples. For SST-2 and MultiRC, we randomly sample 3K examples for fine-tuning. Notably, compared to the 1K example setting used in (Malladi et al., 2023), we sample more data for datasets with larger sizes because we use a much larger batch size.

**Training details.** Following the setup used in (Yue et al., 2023), we adopt a training regime with a learning rate of 2e-5 and a cosine scheduler with a 3% warm-up period, i.e. the learning rate linearly increases from 0 to 2e-5 over the first 3% of training steps, then follows a cosine decay to 0 at the end of training. We set a maximum sequence length of 512. For all experiments on MathInstruct, we standardize the number of gradient steps to correspond to 1K, unless explicitly specified. To simulate a larger batch size, we have also used a gradient accumulation step of 8 in our experiments. We use LoRA with a rank of 128, alpha of 512, and dropout rate of 0.05. For Phi models, we apply LoRA to all attention matrices (i.e. QKV_proj) and two fully connected layers while for Zephyr, we apply LoRA to all attention matrices (i.e. QKVO_proj). All experiments are run on 4 NVIDIA A40 GPUs. We repeat each experiment three times.

**Evaluation datasets.** Following (Yue et al., 2023), we use a variety of popular datasets across both in-domain and out-of-domain datasets. The in-domain datasets include GSM8K (Cobbe et al., 2021), MATH (Hendrycks et al., 2021), and NumGLUE (Mishra et al., 2022). For the out-of-domain

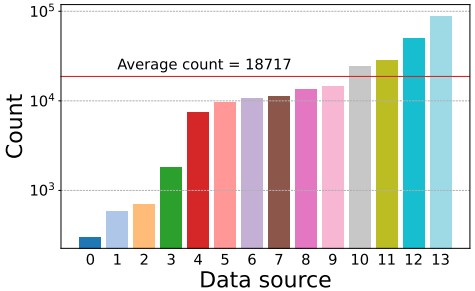 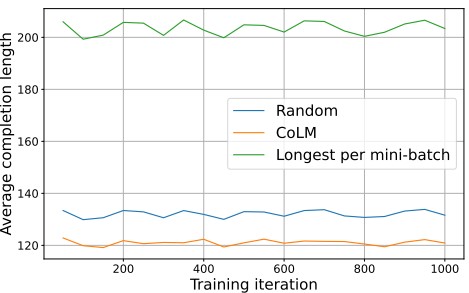

Figure 4: (a) Data distribution of different data sources in MathInstruct. (b) The average completion length of examples selected by CoLMvs. random examples and longest examples in random batches.

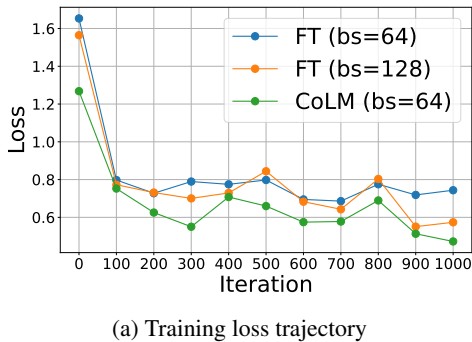

(a) Training loss trajectory

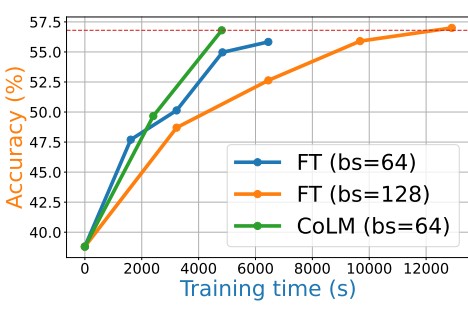

(b) Convergence plot

Figure 5: Fine-tuning Phi-2 on MathInstruct. (a) CoLM yields smallest loss throughout the whole training process; (b) CoLM reaches the optimal performance in less training time.

datasets, we include SVAMP (Patel et al., 2021), Mathematics (Davies et al., 2021), and SimulEq (Koncel-Kedziorski et al., 2016). These datasets collectively cover a wide range of mathematical areas such as algebra, probability, number theory, calculus, and geometry. Furthermore, some questions in these datasets require the application of commonsense, reading comprehension, and multi-step reasoning. All questions are formatted as open-ended.

**Evaluation metric.** We use the standard evaluation metric for open-formed questions, exact match, which measures the model's accuracy by comparing its generated answers against the correct solutions. For an answer to be considered correct, it must match the reference solution precisely. We evaluate under the 0-shot setting with a maximum sequence length of 2048 tokens for decoding. The default prompt is Program-of-Thought (PoT), falling back to Chain-of-Thought (CoT) prompting if the former does not work (Yue et al., 2023).

**Small sources.** Figure 4a visualizes the data distribution of the MathInstruct dataset. It can be seen that the dataset is highly imbalanced with most of the samples belonging to 4 large sources (10 - 13). Therefore, we use a simple heuristic to consider any data sources whose sizes are below the average count as small sources.

## D ADDITIONAL FINE-TUNING RESULTS

### D.1 TRAINING LOSS

For fine-tuning LLMs, downstream performance is generally a better metric than training loss or perplexity. In addition, perplexity does not always correlate with actual task performance on diverse downstream tasks Liu et al. (2023). Furthermore, the MathInstruct dataset doesn't have a held-out validation set to calculate the perplexity. For completeness, we added the training loss in Figure 5a. CoLM consistently yields smallest loss throughout the whole training process.

Table 8: Statistics of pre-training mixture.

Table 9: Downstream acc of pre-trained Llama-60M.

| Dataset | Weight (%) | Small/Large |
|---|---|---|
| EuroParl | 2.8 | Small |
| Github | 28.2 | Large |
| HackerNews | 5.0 | Small |
| NIH Exporters | 3.4 | Small |
| Wikipedia (en) | 60.6 | Large |

| Dataset | PT (bs=128) | PT (bs=256) | CoLM (bs=128) |
|---|---|---|---|
| Squad | 25.4 | 24.8 | **25.9** |
| WiC | 50.2 | 50.7 | **51.6** |
| COPA | 49.0 | 52.0 | **54.0** |
| Avg | 41.5 | 42.5 | **43.9** |

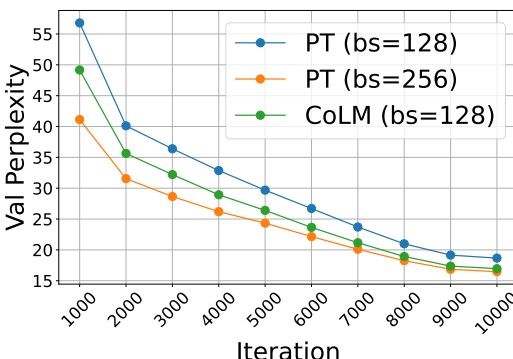

Figure 6: Validation perplexity when pre-training Llama-60M on a mixture of the Pile dataset.

### D.2 CONVERGENCE PLOT IN TERMS OF TRAINING TIME

To highlight the faster convergence rate of CoLM, we plotted Figure 3c with training time as the x-axis. Figure 5b still shows that our method converges faster than normal fine-tuning with both smaller and larger batch sizes.

## E   MEMORY CONSUMPTION

In all our experiments, we used LoRA and a gradient accumulation step of 8, as 4xA40 GPUs did not have enough memory to hold bs=128.

**Memory overhead for CoLM.** The memory overhead of CoLM stems from three main sources the last layer zeroth-order gradient $\hat{g}_{i,t}^{vp}$ in Eq 6, the historical terms of Adam in Eq 8, and the pairwise dissimilarity matrix when solving Eq 8. Because with LoRA, the last layer dimension is 327K and 2560 before and after sparsification and the batch size is 128, the memory overhead is **less than 200MB**. When fine-tuning Phi-2 on MathInstruct with 4xA40 GPUs each with 45G memory (except for bs=256) with LoRA, each GPU can have a maximum device batch-size of 5, so we trained all models with a gradient accumulation step of 8, with LoRA on 4 GPUs. The total batch size = num GPUs x device batch size x gradient accumulation step. Figure 3a illustrates that CoLM (bs=64) outperforms fine-tuning (FT) with bs=256, while requiring 1.8x less memory and being 2.7x faster. Compared to FT bs=128, CoLM requires 20% less memory, while being 30% faster, and obtains 4% higher accuracy.

## F   PRE-TRAINING EXPERIMENTS

**Settings.** We used Llama-60M, which is also used in Zhao et al. (2024b), on a mixture of datasets from the Pile dataset Gao et al. (2020). We selected 5 different datasets *without copyright infringement* including EuroParl, Github, HackerNews, NIH Exporter, and Wikipedia (en). For preprocessing the dataset, we divide each dataset into 1024-token chunks and the statistics are given in Table 8 in which weight means the percentage of 1024-token chunks of each dataset in the mixture. Following Zhao et al. (2024b), we pretrained the model for 10K iterations with a max sequence length of 1024 on 4 GPUs. We also used a gradient accumulation step of 8 similar to fine-tuning

experiments. For evaluation, we calculated the perplexity on a held-out validation set. In addition, we calculated the down-stream accuracy on 3 datasets Squad Rajpurkar (2016), WiC Pilehvar and Camacho-Collados (2018), and COPA Roemmele et al. (2011).

**Validation perplexity.** Figure 6 illustrates the validation perplexity of pre-trained models at different checkpoints during training. CoLM achieves almost the same perplexity as pre-training with 2x batch size.

**Down-stream performance**. Table 9 demonstrates that CoLM improves the downstream accuracy on all 3 evaluation datasets, yielding an improvement of at least 1.4% on average.

