# OpenReview forum: "Mini-batch Coresets for Memory-efficient Language Model Training on Data Mixtures"
_ICLR.cc/2025/Conference — ICLR 2025 Poster_

### Official Review · Reviewer_TWbY · 2024-10-31

**Soundness:** 3
**Presentation:** 3
**Contribution:** 3
**Rating:** 6
**Confidence:** 3

**Summary:**

The paper introduces a method in using smaller mini-batches to simulate the performance of training with larger mini-batches, in the context of Large Language Models (LLMs). In particular, the authors propose coresets for training LLMs (CoLM) which involves the following aspects:

1. Including all examples from small sources in the mini-batch coresets
2. Adapt the mini-batch coreset selection process with ADAM, selecting medoids of big sources.
3. Use zero-order optimization technique based on MeZO in combination with LoRA to estimate the gradients.

Experiments show that the proposed method achieves comparable performance to regular fine-tuning that uses a much larger batch size, while reducing the memory requirements.

**Strengths:**

S1: Well written paper with nice motivation to enable both memory efficiency and performance gains by looking at a data perspective.

S2: Strong experimental results that demonstrate the benefits of their approach compared to baselines. The increase in training time due to mini-batch selection also justified given significant improvement in accuracy.

S3: Ablation studies give insight to the relative importance to each of the three components proposed, with emphasis on gradient approximation, which helps to better understand the benefits of the method.

**Weaknesses:**

W1: There should be some further clarification on how "all" examples of small sources can be included in the mini-batch coreset. Based on experiments, figure 4 shows that the data count for even small data sources can be over 1000 examples. Including all the sources in a single mini-batch does not seem to be plausible, but implied from the methodology and Algorithm 1.

W2: Dependency on having access to the true source data distribution may not always be possible. In these cases, it is difficult to distinguish between small and large sources. In addition, how to determine the boundary between "small" and "large" partitions could be discussed in more detail with experiments.

**Questions:**

Q1: Related to W1 - Is there some sampling or representative subset selection method for the small sources going on behind the scenes?

Q2: Related to W2 - How to determine the boundary between "small" and "large" partitions, especially with skewed distributions where average count may not serve as a good cutoff?

---

> ### Author Response · Authors · 2024-11-23
> **Response to Reviewer TWbY**
>
> We sincerely appreciate the reviewer for acknowledging the clear motivation of our work, our strong experimental results, and the insightful ablation studies.
>
> **W1. Samples from small sources in mini-batches**
>
> As explained in detail in lines 225-237 (also see Line 3 of Algorithm 1 in the appendix), we first randomly draw a mini-batch of samples. Then, the algorithm finds a smaller mini-batch from the examples of the large mini-batch (not from the entire data).
>
> Specifically, we keep all samples of small sources **within the larger mini-batch**. Then, for every big source, we only select its central examples (medoids) in the gradient space from those within the larger mini-batch. That is, centrality is calculated w.r.t the examples within the mini-batch, not the full data.
>
> This is based on our thm 4.2, which shows that: for every big source, central examples w.r.t the larger mini-batch are also central w.r.t the full source, but this does not hold for small sources.
>
> ---
>
> **W2. Distinguish between small and large sources**
>
> We discussed the application of our method to datasets without source info in Sec 5.4. When the data source is unknown or unavailable, one can perform clustering of the target or a smaller proxy model’s hidden states to group data into different “sources”. Then we identify small sources in the same way as if we have the source information (as discussed in lines 238-239). Specifically, having identified c groups, we consider small sources as those with less than |V|/c examples. Table 7 shows our experiments on the SuperGLUE benchmark, and confirms that CoLM effectively improves the performance without any source information as well.
>
> ---
>
> **Q1. Sampling or selection for small sources**
>
> Please refer to our answer in W1.
>
> ---
>
> **Q2. Small and large sources**
>
> Please refer to our answer in W2.
>
> **Q2. Skew distributions**
>
> Note that, as written in Line 177, the ratio of the largest to smallest source size in the MathInstruct dataset is 300, which is sufficiently large. Indeed, the number of small sources is a property of the dataset and may vary depending on the dataset used. Nevertheless, we expect that our strategy which considers any data sources whose sizes are below the average account as small sources might still work in other cases.

---

### Official Review · Reviewer_5bEa · 2024-11-01

**Soundness:** 4
**Presentation:** 4
**Contribution:** 3
**Rating:** 8
**Confidence:** 4

**Summary:**

The paper identifies three key difficulties when applying previous coreset findings methods (methods for finding small mini-batches that have gradients whose mean closely approximates the full gradient) to fine-tuning LLMs: imbalanced data, Adam's update, and large gradient dimensionality. The authors systematically formalize the difficulties and design methods to address them: by treating small-source examples and large-source examples differently, by comparing gradients in the space of Adam updates, and by using zeroth order gradient estimations methods, sparsifying the gradient, and using an appropriate distance metric. The combination of these improvements makes up their method,  Coresets for Training LLMs (CoLM). In their empirical study, authors demonstrate that their method leads to stronger performance compared to a number of baselines (including strong ones) and uses less memory than comparable methods while adding a reasonable overhead in training time.

**Strengths:**

- The paper is very well written. It provides clear motivations of the problem and in thoughtfully structured around the three key difficulties of applying coreset finding methods to LLM fine-tuning. This structure makes the paper clear and very easy to follow.
quality,
- The authors provide extensive ablation analyses about the great majority (if not all) of the improvements they propose, which is greatly appreciated and helps show they are worthwhile.
- The authors empirically demonstrate the effectiveness of CoLM for different pre-trained language models and across many datasets (Math Instruct, SST-2, CB, MultiRC).
- The method often outperforms fine-tuning with twice the batch size.
- The authors demonstrate that the method achieves a smaller memory overhead than a number of representative baselines.

**Weaknesses:**

- Using gradient accumulation + LoRA trivially reduces the memory overhead. The paper would benefit from experiments making an explicit comparison to this case. It seems from Fig 2 (a) that you should be able to claim CoLM is still faster in this case, but not showing it and claiming memory efficiency makes the paper seem weak.

- While the memory overhead is reduced due to the smaller batch size, there is no figure showing how much the total memory decreased by making the batch size smaller when fine-tuning with LoRAs using CoLM. Does the batch size reduction account for a substantial portion of the memory consumption or is the memory consumption dominated by the model parameters given the smaller optimizer states required for one LoRA? Providing a figure with the memory consumption of CoLM compared to fine-tuning at different batch sizes is warranted given the author's claims about memory efficiency.

- Do the methods to estimate gradient distances scale to larger vocabulary sizes? Recent SOTA models (Llama 3 has 128k vocab size whereas phi2,phi3,and zephyr have 50k at most) have been using much larger vocabulary sizes and recent work "Scaling Laws with Vocabulary: Larger Models Deserve Larger Vocabularies" shows this is an important axis along which to scale LLMs. Simply including one experiment with a larger tokenizer 100k vocab+ would strengthen this aspect.

- *I believe the above concerns are easily addressable with small additional experiments. I would be happy to raise my score if these concerns are addressed. *

*List of Typos*:
 "to iteratively selecting small mini-batches from larger random batches." selecting --> select

**Questions:**

(Q1) For Figure 3 (c), how large does the batch size need to be to match your method?

---

> ### Author Response · Authors · 2024-11-23
> **Response to Reviewer 5bEa**
>
> We sincerely thank the reviewer for recognizing the clarity and thoughtful structure of our paper, the extensive ablation analyses, and the strong empirical results across diverse settings.
>
> **W1. CoLM and other memory-efficient methods**
>
> Please note that our method can be easily stacked with LoRA and gradient accumulation while lowering the memory and training time compared to training with large batches. Indeed, all our experiments are done with LoRA (see lines 365-366) and gradient accumulation (clarified in our revision - Training details Appendix B). We also note that (i) LoRA reduces the memory but harms the performance; (ii) gradient accumulation reduces the memory but increases the training time; (iii) our method reduces the memory and training time while considerably improving the performance when applied to full training, or stacked with LoRA and gradient accumulation. A more detailed discussion can be found in our [general comment](https://openreview.net/forum?id=bAFVlpFQvT&noteId=VG5DdiKtk5).
>
> ---
>
> **W2. Memory overhead of CoLM**
>
> Please see our [general comment](https://openreview.net/forum?id=bAFVlpFQvT&noteId=XUYLowYgRo) about actual memory consumption. For Table 1, the activation memory per sample is about 3.9GB while CoLM additional memory is less than 200MB. CoLM effectively reduces the activation memory which scales with the batch size. For example, for fine-tuning Phi-2 with LoRA, CoLM (128->64) requires 20% less memory over bs=128, and CoLM (1024->512) requires 40% memory over bs=1024.
>
> ---
>
> **W3. Larger vocab size & Llama-3**
>
> Our gradient estimation is not affected by the vocab size as we didn’t use the gradient at the lm_embed layer (which is not LoRA fine-tuned) but we used the last V projection matrix (Line 274). In addition, as shown in our [general comment](https://openreview.net/forum?id=bAFVlpFQvT&noteId=XUYLowYgRo), our method is effective for Llama-3 models.
>
> **Typos.** Thanks for pointing it out. We fixed it in our revision.
>
> ---
>
> **Q1. Figure 3c**
>
> We ran normal fine-tuning with bs=512 and it matched the performance of our method. The average performance over 3 seeds is 56.7% $\pm$ 0.3. Please check the new Fig 3c in our revision.

---

> > ### Comment · Reviewer_5bEa · 2024-11-24
> > **The authors have addressed my concerns. I have raised my score.**
> >
> > The authors have addressed my concerns. I have raised my score.

---

> > > ### Author Response · Authors · 2024-11-24
> > > **Thank you for raising your score**
> > >
> > > Thank you for taking the time to read our rebuttal and for raising your score. We greatly appreciate your thoughtful feedback, which helped us improve the quality and clarity of our paper.

---

### Official Review · Reviewer_dq66 · 2024-11-02

**Soundness:** 3
**Presentation:** 2
**Contribution:** 2
**Rating:** 5
**Confidence:** 4

**Summary:**

This paper proposes a method called Coresets for Training Large Language Models (CoLM) to enable memory-efficient training of large language models (LLMs) by constructing mini-batch "coresets." These coresets are small, representative subsets of mini-batches that approximate the gradient of larger batches, allowing effective model training with reduced memory usage. CoLM addresses challenges such as the high memory demands of large mini-batches, the imbalanced distribution in language data, and the complexities of using the Adam optimizer in high-dimensional gradient spaces.

**Strengths:**

- The presentation is clear, and the paper is easy to follow, with only a few minor typos.
- The proposed method, CoLM, is straightforward and demonstrates strong empirical performance.

**Weaknesses:**

- **Limited Base Models**: While the authors mention using Phi-2, Phi-3, and Zephyr, the main results in Tables 1 and 7 only report Phi-2. To more fully demonstrate CoLM’s performance across various models, I recommend including Phi-3 in these tables and consider adding state-of-the-art models, such as LLaMA-3 8B and LLaMA-3.1.

- **Insufficient Discussion of Related Work and Novelty**: One of CoLM’s contributions is gradient normalization for the Adam optimizer, which improves mini-batch approximation by using historical averages. However, previous studies, such as [1], have explored renormalization techniques with mini-batches. It would strengthen the paper if the authors could discuss distinctions between [1] and CoLM to clarify CoLM’s unique contributions.



Reference:

[1] Batch Renormalization: Towards Reducing Minibatch Dependence in Batch-Normalized Models

**Questions:**

- **Computational Complexity**: CoLM introduces additional computational costs to find mini-batches. Could the authors discuss the complexity and time requirements for this estimation algorithm and expand on it in the main paper?

- **Convergence Rate Comparison**: Figure 3 compares CoLM and fine-tuning (FT) by plotting accuracy against training iterations to highlight CoLM’s faster convergence. Could the authors include a comparison of CoLM with LoRA in Figure 3? It would also be valuable to add perplexity or training loss versus iterations to provide further insight into convergence rates.

- **Time vs. Accuracy Comparison**: Given that CoLM involves additional computation to determine mini-batches, comparing accuracy against time would offer a fairer evaluation alongside FT and LoRA in Figure 3(a).

- **Quantitative Memory Cost Analysis**: While the paper claims CoLM uses 4x less memory, it lacks a quantitative breakdown of memory usage. Could the authors add specific memory costs for LoRA, other low-rank adaptation methods, and CoLM in a table? This would provide stronger evidence of CoLM’s memory efficiency.

- **Clarification on Motivation**: 1) In the introduction, the authors state that fine-tuning a model like Phi-2 with 2.7B parameters and a batch size of 128 requires at least 44 GB of GPU memory. However, fine-tuning LLMs primarily incurs memory costs due to model weights and optimizer states, which together cost approximately three times the model weight. Since Phi-2’s model size is under 7 GB, the total memory cost should be around 22 GB rather than 44 GB. Furthermore, batch size typically only affects activation memory and does not significantly impact memory for optimizer states or model weights, and the activation memory cost is not an issue for PEFT methods such as LoRA. 2) **Comparison to LoRA**: LoRA also reduce optimization memory significantly. Besides, LoRA also achieve significantly speedup since it has less trainable parameters. CoLM achieve similar wall clock time with full fine-tuning, but when it applies to full fine-tuning, it will be much slower than LoRA. Could the authors clarify the motivation of the paper from these aspects?



- **Title Consistency**: Please align the submission title, "Memory-efficient Training of Large Language Models with Larger Mini-batches," with the manuscript title, "MINI-BATCH CORESETS FOR MEMORY-EFFICIENT TRAINING OF LARGE LANGUAGE MODELS."

I would like to discuss the questions I raised regarding the weaknesses and concerns with the authors. If my concerns are adequately addressed, I would be willing to reconsider my rating.

---

> ### Author Response · Authors · 2024-11-23
> **Response to Reviewer dq66**
>
> We sincerely appreciate the reviewer for acknowledging the clarity of our paper, as well as the straightforward nature of our proposed method, and its strong empirical performance.
>
> **W1. Limited Base Models**
>
> Figure 2a shows that CoLM outperforms fine-tuning Phi-3 and Zephyr with larger batches on MathInstruct. We report the exact numbers here and have added them to the appendix of our revised version.
>
> | Architectures | bs=64 | bs=128 | CoLM (bs=64) |
> | --- | --- | --- | --- |
> | Phi-3 (4B) | 60.4 ± 0.3 | 61.2 ± 0.6 | 65.4 ± 0.5 |
> | Zephyr-3B (3B) | 37.7 ± 0.6 | 38.0 ± 0.3 | 39.9 ± 0.4 |
>
> To further show the effectiveness of our method across different state-of-the-art models, we ran our method for the LLaMa-3 series and reported the results in our [general comment](https://openreview.net/forum?id=bAFVlpFQvT&noteId=XUYLowYgRo). We see that CoLM consistently outperforms training with small and larger batch sizes, by up to 5.7% and 4% respectively.
>
> ---
>
> **W2. Insufficient Discussion of Related Work and Novelty**
>
> Please note that we use our normalized gradient **only to *select* smaller mini-batches, *not to train* with the normalized gradients**. Indeed, our method *does not interfere with the optimizer*. The optimizer can apply its own normalization, including the gradient normalization introduced by [1] to train on our selected smaller mini-batches. Hence, our contribution is orthogonal to [1].
>
> [1] Batch Renormalization: Towards Reducing Minibatch Dependence in Batch-Normalized Models
>
> ---
>
> **Q1. Computational Complexity**
>
> The additional time for our method stems from three sources: the time to get the zeroth-order gradient of the last layer (Eq 6), the normalized gradient, the sparsified gradient, and solving the submodular facility location optimization problem (Eq 8). The most time-consuming step is getting the zeroth-order gradients, which can be calculated efficiently in only one forward pass as we discussed in Lines 292-296. Figure 2b reports the *wall-clock time including the additional time of the above steps*, and confirms that CoLM is 30% faster than training with bs=128 while using less memory, and outperforms it by 4%. In detail, one iteration of bs=64, bs=128, and CoLM (128->64) in Table 1 cost 3.15, 6.3, and 4.7 (s), respectively.
>
> ---
>
> **Q2. Convergence Rate Comparison**
>
> As mentioned in our experiments section (see lines 365-366 and caption of table 1), in all our experiments including Fig 3, we have already used LoRA for all methods. Please note that LoRA doesn’t make convergence faster, and often harms the final accuracy of the model [2, 3]. In contrast, our method improves the convergence and performance of the model. For fine-tuning, downstream performance is often a better indicator of the model performance than training loss or perplexity. Hence, we reported the downstream accuracy in Fig 3. In our revision, we also added the training loss to Figure 6 in revised Appendix D. At the end of fine-tuning, the training loss of FT (bs=64), FT (bs=128), and CoLM (bs=64) are 0.74, 0.57, and 0.47, respectively. Because the MathInstruct dataset doesn’t have a held-out validation set to calculate the perplexity, we can only calculate the training perplexity. For a fixed-length model, the train perplexity can be approximated by taking the exponential of the training loss. Therefore, the same conclusion holds for training perplexity.
>
> [2] Evans, Paul. "How fast do economics converge?." *Review of Economics and Statistics* 79.2 (1997): 219-225.
>
> [3] Liu, Shih-Yang, et al. "Dora: Weight-decomposed low-rank adaptation." *arXiv preprint arXiv:2402.09353* (2024).

---

> ### Author Response · Authors · 2024-11-23
> **Response to Reviewer dq66 (cont.)**
>
> **Q3. Time vs. Accuracy Comparison**
>
> We plotted Fig 3a with training time as the x-axis and added it to Figure 7 in revised Appendix D. It still shows that our method converges faster than normal fine-tuning with both smaller and larger batch sizes.
>
> ---
>
> **Q4. Quantitative Memory Cost Analysis**
>
> We reported our actual memory in our [general comment](https://openreview.net/forum?id=bAFVlpFQvT&noteId=XUYLowYgRo). CoLM reduces the memory requirement by up to 2x, compared to fine-tuning with *4x batch size*. As discussed in the general comment, The memory overhead of our method is less than 200MB for fine-tuning Phi-2 with LoRA.
>
> **Q5. Clarification on Motivation**
>
> **Memory cost of fine-tuning Phi-2**. Consider an LLM with N billion parameters and full fine-tuning with mixed precision. The optimizer maintains a copy of the model in FP32 precision, consuming 4N memory. The gradients, momentum, and second-moment vectors are all stored in FP32 precision with each requiring 4N memory. Consequently, the total memory required is at least 16N = 43.2 GB if N = 2.7 for Phi-2.
>
> **Activation cost for LoRA**. LoRA only reduces the optimizer state memory but increases the model weights and activation memory, by adding additional low-rank matrices. The activation memory still scales with larger device batch sizes.
>
> **Stack CoLM with other memory-efficient methods**. We note that our method is not a replacement for LoRA and other memory-efficient methods and can be easily stacked with them to further reduce the memory requirements. Indeed, as mentioned in lines 365-366 and the caption of table 1, all our experiments are done with LoRA. Fig 2b shows that when stacked with LoRA, CoLM is indeed much faster than training with the larger batch. Fig 2b shows that our method (CoLM 128->64) is actually 30% faster than normal fine-tuning with a batch size of 128. Notably, unlike existing memory-efficient training methods that harm accuracy, our method outperforms training with bs=128 by around 4%!
>
> ---
>
> **Q6. Title Consistency**
>
> Thanks for pointing this out. We will change the submission title in the final version.

---

> > ### Comment · Reviewer_dq66 · 2024-11-24
> >
> > I appreciate the authors' response and thank you for the discussion. I have more questions I’d like to discuss further:
> >
> > In Section 4.2, the authors discuss the method for obtaining low-dimensional gradient estimates, which appears similar to Espace and Galore [1,2]. However, the differences between the proposed method and [1,2], as well as the advantages of the method in Section 4.2 compared to [1,2], are not clearly addressed. The citation for [1] is missing. Could the authors further discuss on how their approach differs with [1,2]?
> >
> > References:
> > [1] Espace: Dimensionality Reduction of Activations for Model Compression
> > [2] Galore: Memory-Efficient LLM Training by Gradient Low-Rank Projection

---

> > > ### Author Response · Authors · 2024-11-25
> > > **Response to Reviewer dq66**
> > >
> > > Thank you for reviewing our rebuttal and providing a follow-up question. Please note that [1] was published on arXiv on October 7, 2024, after the ICLR 2025 submission deadline. In fact, our contributions are orthogonal to [1, 2] as we discuss below.
> > >
> > > CoLM improves convergence and performance of LLMs by finding smaller mini-batches that closely match the gradient of larger mini-batches. In contrast, [1, 2] directly change the optimizer. Importantly, CoLM does not interfere with the optimization process and thus can be applied to any memory-efficient optimizer including [1, 2, LoRA, etc].
> > >
> > > CoLM uses sparsified zeroth-order gradient estimates to find the smaller mini-batches. Finding low-dimensional gradient estimates is just *one component* of CoLM. Our other contributions include balancing sources in the small mini-batch, and selecting smaller mini-batches by matching the normalized (instead of vanilla) gradients. Next, we discuss why [1, 2] cannot be used to find lower-dimensional gradient estimates to find smaller mini-batches.
> > >
> > > - Espace reduces the dimensionality of activations (to speed up inference and fine-tuning) and leaves the weight matrices untouched. As the dimensionality of the gradient is the same as the weight dimensionality, Espace cannot be applied to find lower-dimensional gradients. Espace also increases the memory requirement by introducing an additional matrix P (first paragraph of page 5 in [1]). In contrast, CoLM only requires one forward pass to calculate low-dimensional zeroth-order gradient estimates and does not need to store additional parameters.
> > > - GaLore reduces the *optimizer state memory* by leveraging the slow-changing low-rank structure of the gradient of the weight matrix. GaLore does not approximate the weight matrix itself as low-rank (page 2 of [2]). Hence, it does not reduce the dimensionality of the gradients. Besides, due to the additional cost of SVD, GaLore is 25% slower than normal training (c.f. Table 11 in Appendix E [2]). In contrast, CoLM speeds up training with LoRA by 30% (Fig 2b).
> > > - Importantly, unlike the memory-efficient methods including [1, 2, LoRa] that harm the performance, CoLM improves the performance.
> > >
> > > Generally, CoLM can use any existing technique applicable to calculate lower-dimensional gradient estimates. In our experiments (Table 5), we showed that our sparsified zeroth-order gradients outperform low-dimensional gradient similarity estimation using random projections as well as sparsified first-order gradients.
> > >
> > > We also conducted new experiments to apply SVD to lower the dimensionality of our zeroth-order last-layer gradients (Eq 6). Following the fine-tuning setting of GaLore, we used rank r = 8 and subspace change frequency T = 200. Because the dimension of last LoRA layer is 128 x 2560, after projection, the gradient dimensionality becomes r * n = 8 * 2560 = 20480. Table below details the results of fine-tuning Phi-2 on MathInstruct.
> > >
> > > | Gradient estimate | In-domain | Out-domain | Avg |
> > > | --- | --- | --- | --- |
> > > | Low-rank  | 51.0 ± 0.2 | 58.1 ± 0.8 | 54.6 ± 0.4 |
> > > | Sparsified | 51.9 ± 0.3 | 61.4 ± 1.6 | 56.6 ± 0.9 |
> > >
> > > We see that our sparsified gradient estimates outperform the low-dimensional gradients found via SVD. We have added the new experimental results to our revised Appendix G.
> > >
> > > [1] Espace: Dimensionality Reduction of Activations for Model Compression
> > >
> > > [2] Galore: Memory-Efficient LLM Training by Gradient Low-Rank Projection

---

> ### Author Response · Authors · 2024-12-01
> **Response to Reviewer dq66**
>
> Thank you for your response. We believe your concerns are already addressed in our original submission and rebuttal. We will iterate over these points below:
>
> Regarding our method, point (2) is not accurate, as we “do not revise the Adam optimizer”. As discussed in Sec 4.2 (e.g. see line 257: “For *selecting mini-batch coresets* for training with Adam …”), and our response to your review (see [W2](https://openreview.net/forum?id=bAFVlpFQvT&noteId=0as9T8gIVd)), our method does not change or revise the Adam optimizer. We only use the normalized gradient to **select the subset**, not to train with them. In fact, (2) and (3) in addition to (1) allow “finding the subsets”.
>
> 1. Please see Lines 365-366 in our experiments under the “training details header”, where the first sentence mentions that “our experiments use LoRA”. This is also clearly mentioned in the caption of Table 1, where the first sentence mentions that “Accuracies on in-domain and out-of-domain datasets when fine-tuning Phi-2 **with LoRA**”. We’ll highlight this in the caption of the figures too.
> 2. The one-shot baselines are already briefly discussed in our related work section (lines 108-110), where we mentioned middle perplexity as the most successful one-shot method. However, as discussed there, such baselines are most useful for *pretraining*, not fine-tuning LLMs. The one-shot baselines are further listed and cited under the “baselines header” in our experiments section (lines 369-374). In fact, the one-shot methods mentioned by the reviewer are already the baselines we compared to in our Table 1, and confirmed the clear advantage of our method! It is also mentioned in the caption of Table 1, that “we compare to one-shot methods”: CL (completion length), BL (big loss), GN (gradient norm), LC (low confidence), FL or SO (submodular facility location), MP (middle perplexity). Consistent with the discussion in our related work section, MP is the best one-shot baseline and CoLM outperforms MP by 3%.
> 3. Please see line 365 of our experiments section, the first line of the “training details header” mentions: “We use LoRA with a rank of 128, alpha of 512”. This is even mentioned in our introduction (see line 79 which mentions that we use LoRA in our experiments.). Please note that alpha in LoRA doesn’t affect the memory consumption of LoRA. For precision, we used the default precision of each pre-trained model, i.e. float16 for Phi-2, bfloat16 for Phi-3, Zephyr-3B, and Llama-3. We will add this to our version.
> 4. Please note that there is no memory saving corresponding to different parts of our method. As discussed in our [general comment](https://openreview.net/forum?id=bAFVlpFQvT&noteId=VG5DdiKtk5), *our method reduces the activation memory* by selecting smaller mini-batches. Without part (3) which reduces the gradient dimensionality, one cannot select a subset (this is discussed in lines 183-187 before we discuss our method, and is iterated over in lines 268-269 before we start Sec 4.3): “This is because in such a high dimensional space, distances become vacuous”. Our Table 5 shows other ways of finding low-dimensional gradient estimates to enable subset selection. Again, note that (1-3) are all *to select a subset* not to change the optimizer or reduce the weight or optimizer memory, etc. The total *memory overhead* of our method is less than 200MB, from which 82.5MB is due to using historical terms in part (2) and the rest is due to part (3). Thus, one can use CoLM without (2) to slightly reduce the memory overhead but trading off 1% of the performance (see Table 2). (1) doesn’t introduce any overhead. While the total memory overhead of our method is negligible, per reviewer’s suggestion, we’ll add the above numbers to our revision.
>
> 5. As discussed in our [general comment](https://openreview.net/forum?id=bAFVlpFQvT&noteId=VG5DdiKtk5), our method reduces the batch size and thus reduces the *activation memory*, not the optimizer memory. Hence, it is complementary to LoRA and other parameter efficient methods. In our [general comment cont.](https://openreview.net/forum?id=bAFVlpFQvT&noteId=XUYLowYgRo), we discussed in detail how much memory CoLM saves over LoRA, under different settings (this is also added to our revised version). Note that all the experiments in our rebuttal are all done with LoRA and show the memory saving over LoRA. Particularly, for larger batches, our method reduces the (activation) memory requirement by around 2x, and this includes the 200MB overhead of our method. Notably, our method improves the performance, while LoRA and other parameter-efficient methods harm the performance.
>
> Given that most of your concerns were already addressed in our original submission or rebuttal, we do hope that you consider giving our paper stronger support. We believe our work is complete, and our novel contributions enable training LLMs with larger batch size and superior performance. Thus, our work deserves more than a borderline score.

---

### Official Review · Reviewer_AKmp · 2024-11-08

**Soundness:** 4
**Presentation:** 4
**Contribution:** 3
**Rating:** 8
**Confidence:** 4

**Summary:**

The paper introduces CoLM (Coresets for Training LLMs), a method for memory-efficient training of large language models (LLMs) using mini-batch coresets that approximate the gradients of larger batches. CoLM addresses specific challenges in LLM training, such as data imbalance, high gradient dimensionality, and compatibility with the Adam optimizer. CoLM addresses them by including all examples from smaller data sources, normalizing gradients for Adam, and using zeroth-order gradient estimates. It reduces memory requirements by 2x while maintaining or surpassing performance compared to larger mini-batches.

**Strengths:**

1. The approach takes a unique and innovative perspective on memory-efficient training. Unlike typical methods in field, it focuses on data selection and reduces batch size to achieve memory-efficient training.
2. The paper is clear to understand and well motivated.
2. The paper presents comprehensive experiments on diverse datasets and models, substantiating the benefits of CoLM over various baselines.

**Weaknesses:**

1. Computational overhead: Figure 2(b) suggests that the computational overhead for CoLM is notably high, doubling the training time in some cases, whereas typical memory-efficient methods usually incur 5-20% overhead. A comparison of "memory vs. computational overhead vs. performance" with other methods would be beneficial.
2. The paper mainly evaluates CoLM in a fine-tuning context, where model and optimizer state memory are the main bottlenecks. It would be valuable to see CoLM tested in pre-training or continual pre-training settings, where activation memory dominates due to large batch sizes. This could pose additional challenges given complex data structures and extensive amount of training data.
3. Not enough details of algorithm and discussion on hyper-parameters.

**Questions:**

1. Is zeroth-order necessarily needed? How about estimating using standard first-order methods?
2. What hypeperparametrs involved in CoLM? Like how many small sources and large sources needed? How much is the cost of tuning those hyper parameters? Are they sensitive to different data structures and types of data?

---

> ### Author Response · Authors · 2024-11-23
> **Response to Reviewer AKmp**
>
> We sincerely thank the reviewer for recognizing our innovative approach, the strong motivation of our work, as well as our comprehensive experiments across diverse datasets and models.
>
> **W1. Computational overhead**
>
> There might be a misread of our Fig 2b. Our method (CoLM) with bs=64 simulates and outperforms training with bs=128, using lower memory requirements. Hence, our method should be compared with bs=128. Fig 2b shows that **(CoLM) is actually 30% faster than normal fine-tuning with bs=128**. Notably, unlike existing memory-efficient training methods that harm accuracy, our method outperforms training with bs=128 by around 4%! Besides, our method can be easily stacked with existing memory-efficient methods. Indeed, as mentioned in the Experiments section (line 365 and Table 2), all our experiments were done with LoRA and improve the memory efficiency over LoRA.
>
> ---
>
> **W2. Pre-training experiments**
>
> Our method mimics training with larger batch sizes (by matching their gradient), and thus can deal with complex and big datasets, as normal training with larger batch sizes does.
>
> Indeed, for pretraining when the batch size is larger, the activation memory dominates the optimizer state and model weight memory. Thus, as discussed in our [general comment](https://openreview.net/forum?id=bAFVlpFQvT&noteId=XUYLowYgRo), ****CoLM reduces the memory requirement even further when batch size is larger****.
>
> **Pre-training experiments**. To demonstrate the effectiveness of our method in the pre-training setting, we pre-trained Llama-60M (used in [1]) on a mixture of datasets from the Pile dataset [2]. We selected 5 different datasets **without** **copyright infringement** including EuroParl, Github, HackerNews, NIH Exporter, and Wikipedia (en). For preprocessing the dataset, we divide each dataset into 1024-token chunks, and the statistics are given in the table below.
>
> | Dataset | Size | Small or Large source |
> | --- | --- | --- |
> | EuroParl | 2.8% | Small |
> | Github | 28.2% | Large |
> | HackerNews | 5.0% | Small |
> | NIH Exporter | 3.4% | Small |
> | Wikipedia (en) | 60.6% | Large |
>
> Following [1], we pretrained the model for 10K iterations with a max sequence length of 1024 on 4 GPUs. For evaluation, we calculated the perplexity on a held-out validation set. In addition, we calculated the downstream accuracy on 3 datasets Squad, WiC, and COPA.
>
> In terms of validation perplexity (cf. Figure 9 in revised Appendix F), CoLM achieves a similar value as FT (bs=256), which is lower than FT (bs=128) at the end of training.
>
> The following table demonstrates that CoLM improves the downstream accuracy
>
> on all 3 evaluation datasets, yielding an improvement of at least 1.4% on average.
>
> | Method | Squad | WiC | COPA | Avg |
> | --- | --- | --- | --- | --- |
> | bs=128 | 25.4 | 50.2 | 49.0 | 41.5 |
> | bs=256 | 24.8 | 50.7 | 52.0 | 42.5 |
> | CoLM (256->128) | 25.9 | 51.6 | 54.0 | 43.9 |
>
> We believe that this is a great addition to our work, and we will add larger pre-training experiments to our final version. Thank you for the suggestion!
>
> [1] Zhao, Jiawei, et al. "Galore: Memory-efficient llm training by gradient low-rank projection." *arXiv preprint arXiv:2403.03507* (2024).
>
> [2] Gao, Leo, et al. "The pile: An 800gb dataset of diverse text for language modeling." *arXiv preprint arXiv:2101.00027* (2020).
>
> ---
>
> **W3. Algorithm & Hyper-parameters**
>
> Our only hyper-parameter is the sparsity level. As we showed in Table 4, using the same number of dimensions as that of the model’s last hidden state yields the best results. Thus, in practice one does not need to tune this hyper-parameter. The pseudo-code of our algorithm is illustrated in revised Appendix C.

---

> ### Author Response · Authors · 2024-11-23
> **Response to Reviewer AKmp (cont.)**
>
> **Q1. Zeroth-order vs First-order gradient**
>
> As discussed at the beginning of Sec 4.3 (line 272), the high-dimensional gradients of LLMs are very noisy. The zeroth-order gradients have two benefits: (1) they are smoother than actual gradients calculated with backprop, hence allowing finding higher-quality subsets (see our ablation study in Table 5 that confirms that using zeroth-order gradients outperform standard first-order gradients); and (2) as discussed in lines 293-298, they can be calculated quickly using just one forward pass in a memory efficient manner (which is our objective).
>
> ---
>
> **Q2. Hyper-parameters**
>
> Our only hyper-parameter is the sparsity level. Our ablation study in Table 4 showed the dimensionality of the last hidden state works well, and hence in practice one does not need to tune this parameter.
>
> **Q2. Data source & data structure**
>
> NLP datasets are usually a mixture of data from different sources. For example, the MathInstruct data contains 14 sources, including TheoremQA, Camel-Math, College-Math, etc. Thus, the number of small sources is a property of the dataset and is not a parameter to tune by our method. As discussed in lines 238-239, we simply consider any data sources whose sizes are below the average account as small sources. We expect our method to work on datasets with various numbers of sources and also on datasets without any specific sources, as we confirmed in Sec 5.4 (see Table 7), on the SuperGLUE benchmark without specific sources.

---

> > ### Comment · Reviewer_AKmp · 2024-11-24
> >
> > I would like to raise my score as the authors has addressed my concerns and questions.

---

> > > ### Author Response · Authors · 2024-11-24
> > > **Thank you for raising your score**
> > >
> > > Thank you for reading our rebuttal and raising your score. Your constructive feedback was instrumental in helping us improve the clarity and quality of our paper.

---

### Author Response · Authors · 2024-11-23
**General Comment**

We thank all the reviewers for their valuable feedback and supporting our work. In our general comment, we discuss the memory and time requirement of our method CoLM and when it yields memory reduction, speedup, and performance improvement in different settings: (i) stacked with LoRA and gradient accumulation, (ii) large batch size, (iii) applied to state-of-the-art models (Llama3). We have added the discussion to our revised version.

**CoLM Effectively Reduces the Activation Memory.** The memory required for training an LLM can be decomposed into three parts: activation memory + weight memory + optimizer state memory. Memory-efficient methods often reduce the weight or optimizer-state memory. For example, LoRA reduces the optimizer state memory but slightly increases the weight and activation memory by adding low-rank matrices. Orthogonal to such methods, CoLM effectively reduces the activation memory by reducing the batch size. Hence, stacked with memory-efficient methods such as LoRA, it can further reduce memory, particularly when batch size is large. Notably, while memory-efficient methods harm the performance, CoLM effectively improves the performance over training with small and larger batches. For reference, full-parameter fine-tuning of Phi-2 on MathInstruct with bs=128 achieves 56% average accuracy, while LoRA fine-tuning with bs=128 achieves 52.6%, and CoLM (128->64) with LoRA achieves 56.5%.

**CoLM Easily Stacks with LoRA and Gradient Accumulation**. In all our experiments in the paper, we used LoRA (lines 365-366 and caption of table 1), and a gradient accumulation step of 8 for all methods, as our A40 GPUs didn’t have enough memory to hold bs=128 (we added the detailed explanation to our revised Appendix E). As explained above, CoLM easily stacks with existing memory-efficient methods.

Below, we first discuss the negligible memory overhead of CoLM and then clarify its benefit in reducing the memory requirement, speeding up the training, and improving the performance in different settings.

**Memory overhead of CoLM**. The memory overhead of CoLM stems from three main sources the last layer zeroth-order gradient $\hat{g}_{i,t}^{vp}$ in Eq (6), the historical terms of Adam in Eq (8), and the pairwise dissimilarity matrix when solving Eq (8). Because with LoRA, the last layer dimension is 327K and 2560 before and after sparsification, for CoLM (128->64), the memory overhead is less than 200MB.

The following table summarizes the memory, wall-clock time, and average accuracy, when fine-tuning Phi-2 on MathInstruct (setting of Table 1), with 4xA40 GPUs each with 45G memory. Using LoRA, each GPU can have a maximum device batch size of 5, so we trained all models with a gradient accumulation step of 8, with LoRA on 4 GPUs. The total batch size = #GPUs x device batch size x gradient accumulation step.

---

> ### Author Response · Authors · 2024-11-23
> **General Comment (cont.)**
>
> **CoLM Reduces the Memory Requirement and Obtains a Higher Accuracy, when Applied with (i) Smaller Gradient Accumulation Steps, or (ii) Larger Batch Sizes**
>
> **(i) Smaller Gradient Accumulation Steps.** The following table shows that CoLM (bs=64) outperforms fine-tuning (FT) with bs=256 while requiring 1.8x less memory and being 2.7x faster. Compared to FT bs=128, CoLM requires 20% less memory, while being 30% faster, and obtains 4% higher accuracy.
>
> | Method | LoRA | Gradient accumulation step (GAS) | Memory per each of the 4 GPUs (GB) | Time (h) | Avg acc |
> | --- | --- | --- | --- | --- | --- |
> | FT bs=64 | Yes | 8 | 28.1 | 0.9 | 50.1 ± 0.2 |
> | FT bs=128 | Yes | 8 | 35.9 | 1.8 | 52.6 ± 0.6 |
> | FT bs=256* | Yes | 8 | 51.6 | 3.6 | 55.3 ± 0.5 |
> | CoLM (bs=128->64) | Yes | 8 | 28.3 | 1.3 | 56.5 ± 0.9 |
> *bs=256 is run on 4xH100
>
> Gradient accumulation linearly increases the training time. With larger GPU memory (H100 94GB), one can use no gradient accumulation (step = 1) to significantly speed up training. In this case, the activation memory dominates the model weight and optimizer state memory (even when training with LoRA). Therefore, CoLM can reduce the memory consumption over bs=128 by around 2x.
>
> | Method | Gradient accumulation step (GAS) | Estimated memory per each of the 4 GPUs (GB) |
> | --- | --- | --- |
> | bs=64 | 1 | 82.8 |
> | bs=128 | 1 | 145.3 (OOM) |
> | CoLM (bs=64) | 1 | 83.0 (2x less memory than bs=128) |
>
> **(ii) Larger Batch Sizes.** When the batch size is larger, the activation memory dominates the optimizer state and weight memory. The following table shows how the estimated memory requirement (per GPU) of CoLM for fine-tuning Phi-2 increases when using larger batch sizes (also shown in Figure 8 in our revised Appendix E). Notably, with 4xA40 and using gradient accumulation step=8 with LoRA, for bs=2048, CoLM (bs=1024) will require almost 2x less memory than fine-tuning with bs=2048. A larger batch size is useful in particular for pretraining. Our [new experiments](https://openreview.net/forum?id=bAFVlpFQvT&noteId=hx2pEO8y7O) for pre-training Llama on Pile, confirm the benefits of CoLM to pretraining.
>
> | batch size (bs) | 64 | 128 | 256 | 512 | 1024 | 2048 |
> | --- | --- | --- | --- | --- | --- | --- |
> | FT (GB), GAS=8 | 28.1 | 35.9 | 51.6 | 82.8 | 145.3 | 270.3 |
> | CoLM (GB), GAS=8 | 24.3 | 28.3 | 36.2 | 52.2 | 84.4 | 149.7 |
>
> **(iii) Llama-3 experiments**. To demonstrate the effectiveness of our method to the recent state-of-the-art models with larger vocabulary size, we fine-tuned Llama-3 models with LoRA using the same setting as Phi-2 on MathInstruct. We report the average accuracy on in- and out-of-domains below. We see that CoLM with bs=64 outperforms normal fine-tuning with bs=128, with smaller memory requirements.
>
> | Architectures | bs=64 | bs=128 | CoLM (bs=64) |
> | --- | --- | --- | --- |
> | Llama-3.2-1B | 20.2 ± 0.2 | 20.3 ± 0.1 | 22.0 ± 0.4 |
> | Llama-3.2-3B | 35.1 ± 0.1 | 36.1 ± 0.0 | 37.7 ± 0.5 |
> | Llama-3.1-8B | 45.4 ± 0.5 | 47.1 ± 0.7 | 51.1 ± 0.3 |

---

### Meta-Review · Area_Chair_MfKM · 2024-12-22

**Metareview:**

This paper introduces CoLM, a memory-efficient method that constructs small, representative mini-batch coresets to approximate the gradients of larger batches. CoLM addresses three key challenges in applying coreset methods to LLMs: (1) imbalanced data distributions, (2) the complexities of Adam's update rule in high-dimensional gradient spaces, and (3) computational difficulties due to large gradient dimensionality.

The proposed method tackles these issues by including all examples from smaller data sources, adapting gradient comparisons to the Adam update space, and employing zeroth-order optimization techniques with sparsified gradients. CoLM achieves comparable or superior performance to standard fine-tuning methods with large mini-batches while reducing memory requirements and adding only modest computational overhead.

While this paper has several strengths, **it also has several weaknesses**, as pointed out by Reviewer dq66. (1) The three key components of the proposed method have been explored in prior work, and the paper lacks detailed comparisons and explanations to highlight its unique contributions. (2) The main results are based on fine-tuning with LoRA, which inherently has low memory requirements; demonstrating a 2x memory reduction would be more compelling in the context of full fine-tuning. (3) Appropriate ablation studies are needed to evaluate the contributions of each component independently.

Despite these limitations, the paper’s strengths outweigh its weaknesses. Most reviewers supported acceptance, with several providing high scores, reflecting confidence in the paper’s contributions. Therefore, I recommend accepting this paper.

**Additional Comments On Reviewer Discussion:**

During the rebuttal period, most reviewers maintained a positive score on the paper. Some reviewers raised concerns about the experimental setup, scale, and choice of models, which the authors addressed in their response. However, these clarifications did not significantly change the reviewers' opinions. While Reviewer dq66 maintained a borderline position, the other reviewers leaned toward acceptance. Overall, the consensus was in favor of accepting the paper.

---

### Decision · Program_Chairs · 2025-01-22

Accept (Poster)